# Multiple Physics Pretraining for Spatiotemporal Surrogate Models

**Michael McCabe**[*,1,2]**, Bruno Régaldo-Saint Blancard**[1]**, Liam Parker**[1]**, Ruben Ohana**[1]**,
Miles Cranmer**[3]**, Alberto Bietti**[1]**, Michael Eickenberg**[1]**,Siavash Golkar**[1]**,
Geraud Krawezik**[1]**, Francois Lanusse**[1,4]**, Mariel Pettee**[1,5]**,
Tiberiu Tesileanu**[1]**, Kyunghyun Cho**[6,7,8]**, Shirley Ho**[1,6,9]

The Polymathic AI Collaboration

[1] Flatiron Institute, [2] University of Colorado Boulder, [3] University of Cambridge,
[4] Université Paris-Saclay, Université Paris Cité, CEA, CNRS, AIM,
[5] Physics Division, Lawrence Berkeley National Laboratory, [6] New York University,
[7] Prescient Design, Genentech, [8] CIFAR Fellow, [9] Princeton University

## Abstract

We introduce multiple physics pretraining (MPP), an autoregressive task-agnostic pretraining approach for physical surrogate modeling of spatiotemporal systems with transformers. In MPP, rather than training one model on a specific physical system, we train a backbone model to predict the dynamics of multiple heterogeneous physical systems simultaneously in order to learn features that are broadly useful across systems and facilitate transfer. In order to learn effectively in this setting, we introduce a shared embedding and normalization strategy that projects the fields of multiple systems into a shared embedding space. We validate the efficacy of our approach on both pretraining and downstream tasks over a broad fluid mechanics-oriented benchmark. We show that a single MPP-pretrained transformer is able to match or outperform task-specific baselines on all pretraining sub-tasks without the need for finetuning. For downstream tasks, we demonstrate that finetuning MPP-trained models results in more accurate predictions across multiple time-steps on systems with previously unseen physical components or higher dimensional systems compared to training from scratch or finetuning pretrained video foundation models. We open-source our code and model weights trained at multiple scales for reproducibility.

## 1 Introduction

In recent years, the fields of natural language processing and computer vision have been revolutionized by the success of large models pretrained with task-agnostic objectives on massive, diverse datasets (Chen et al., 2020; Devlin et al., 2018; He et al., 2021). This has, in part, been driven by the development of self-supervised pretraining methods which allow models to utilize far more training data than would be accessible with supervised training (Balestriero et al., 2023). These so-called "foundation models" have enabled transfer learning on entirely new scales. Despite their task-agnostic pretraining, the features they extract have been leveraged as a basis for task-specific finetuning, outperforming supervised training alone across numerous problems especially for transfer to settings that are insufficiently data-rich to train large models from scratch (Bommasani et al., 2021).

---

[*]Contact: `mmccabe@flatironinstitute.org`

38th Conference on Neural Information Processing Systems (NeurIPS 2024).

Deep learning for computational science has begun to see first steps in this direction. Large domain-specific pretrained models have emerged in diverse fields such as chemistry (Bran et al., 2023; Chithrananda et al., 2020), medicine (Jiang et al., 2023; Tu et al., 2023), astrophysics (Leung & Bovy, 2023; Nguyen et al., 2023b), and climate (Nguyen et al., 2023a) and the trend only seems to be growing as more and more models are developed for new fields both as refined versions of existing large language models and as new models trained entirely on field-specific data.

In this work, we demonstrate that similar approaches can be extended to the surrogate modeling of spatiotemporal physical systems. Spatiotemporal prediction tasks, like those found in fluids, solids, or general continuum mechanics, have attracted significant attention from the deep learning community. From direct prediction methods (Dang et al., 2022; Li et al., 2020; Lusch et al., 2018; Pfaff et al., 2021; Stachenfeld et al., 2022) to neural PDE solvers (Bruna et al., 2022; Raissi et al., 2019), researchers have sought to develop fast, accurate models for physics either as faster surrogates for the partial differential equation (PDE) solvers that dominate the field or to simulate systems that cannot be exactly described or resolved by current mechanistic models and available hardware. While directly outperforming PDE solvers is difficult (Grossmann et al., 2023), deep learning has already begun to impact fields like atmospheric science (Ben-Bouallegue et al., 2023; Bi et al., 2023; Lam et al., 2023; Pathak et al., 2022) and cosmology (Cranmer et al., 2021; He et al., 2019; Jamieson et al., 2023), where the systems are too large or too imprecisely described to be simulated exactly.

Unfortunately, outside of a few observation-rich outliers, settings where numerical simulation is expensive or unreliable also tend to be settings where the difficulty of acquiring training data makes it impractical to train surrogates conventionally. Most deep learning-based surrogates thus far have focused on specific physical systems or families of parameterized PDEs where data can easily be acquired. However, for the low-data settings often found in simulation-driven exploration and design, it would be valuable to have large, task-agnostic models with a broad understanding of common physical behavior to act as a foundation for finetuning.

**Contributions.** To address this need, we introduce *Multiple Physics Pretraining* (MPP), a new approach for task-agnostic pretraining of physical surrogate models. Our method enables large-scale pretraining for transfer across diverse physics which we study using fluid-oriented benchmarks. Our specific contributions are:

- We develop MPP, a pretraining approach in which we embed multiple hetereogeneous physical systems into a shared embedding space and learn to autoregressively predict the dynamics of all systems simultaneously.

- We show that single transformer models pretrained with MPP are able to match or surpass modern task-specific baselines without applying task-specific finetuning to the MPP models.

- We demonstrate the transfer capabilities of models trained with MPP to systems with limited training examples (referred to as *low-data systems* thereafter) displaying new physics in the form of previously unseen parameter regimes generating notably different qualitative behavior and inflated to higher dimensions.

- We open-source our code and provide our pretrained models at a variety of sizes for the community to experiment with on their own tasks.

## 2 Background

**Notation.** Let $S$ be an arbitrary physics-driven spatiotemporal dynamical system, either described by a parameterized family of PDEs with fixed parameters, or where snapshots are gathered from observation of a unique physical phenomenon. To simplify notation, we discuss systems with a single state variable in one spatial dimension. A continuous state variable for system $S$ is represented as $u^S(x,t) : [0, L_S] \times [0, \infty) \to \mathbb{R}$. We discretize the system uniformly in space and time at resolutions $N_S$, $T_S$ respectively. A snapshot $\boldsymbol{u}_t^S \in \mathbb{R}^{N_S}$ represents the value of state variable $u^S$ at all $N_S$ spatial discretization points at time $t$. Our pretraining task is then to learn a single model $\mathcal{M}$ that can take a uniformly spaced sequence of $T_S$ snapshots $\boldsymbol{U}_t^S = [\boldsymbol{u}_{t-T_s \Delta t_S}^S, \ldots, \boldsymbol{u}_t^S]$ from system $S$ sampled from some distribution over systems and predict $\mathcal{M}(\boldsymbol{U}_t^S)$ such that $\mathcal{M}(\boldsymbol{U}_t^S) \approx \boldsymbol{u}_{t+\Delta t_S}^S$.

**Autoregressive Pretraining.** In vision and language, the dominant pretraining strategies include autoregressive prediction (Radford et al., 2018), masked reconstruction (Devlin et al., 2018; He

et al., 2021), and contrastive learning (Chen et al., 2020). In language, autoregressive generation emerged as a convenient self-supervised task. In surrogate modeling of dynamical systems, next-step prediction is often a primary goal. This makes autoregressive pretraining a natural choice of objective for training time-dependent surrogate models.

We note that it is common to use the simulation parameters to condition the predictions of models operating on PDE-generated data (Gupta & Brandstetter, 2022; Subramanian et al., 2023; Takamoto et al., 2023). Our goal is not to develop a new PDE solver, but rather to design an approach that is broadly applicable to both observed and simulated dynamics. Thus, we do not assume a known functional form in MPP and the model must instead implicitly infer the impact of these parameters on the dynamics from the history provided in $U_t^S$.

**Surrogate Modeling for Spatiotemporal Physical Systems.** We are primarily concerned with modeling dynamical systems varying in both time and space, where the time evolution of the system is intrinsically tied to spatial relationships amongst the state variables according to physical laws. PDEs are one of the primary modeling tools for this setting. They are often derived from fundamental conservation laws of properties such as mass, momentum, and energy (Farlow, 1993). Many PDEs describe variations of the same physical laws, which is why concepts like diffusion, advection, reactivity, and connections between time and spatial gradients appear in many different PDEs. These shared underlying principles suggest we can extract features relevant to multiple physical systems.

# 3 Related Work

**Foundation models.** Massive pretrained models dubbed "foundation models" (Bommasani et al., 2021), particularly large transformer-based architectures (Vaswani et al., 2017), have recently attracted significant attention. The most prevalent foundation models are pretrained language models like GPT (Brown et al., 2020; Radford et al., 2018, 2019) and BERT (Devlin et al., 2018). Emergent abilities (Wei et al., 2022) demonstrated by large language models highlight the importance of scale in manifesting higher-order capabilities absent at smaller scales. Vision has seen similar developments with the growth of masked (He et al., 2021; Tong et al., 2022) and contrastive (Chen et al., 2020) pretraining. The data in this work is insufficiently diverse to call the resulting models "foundational". However, we provide the first large-scale implementation of successful multiple nonlinear physics pretraining for spatiotemporal systems.

**Scientific machine learning.** While a wide range of architectures have been employed for physical surrogate modeling (Bar & Sochen, 2019; Han et al., 2018; Sirignano & Spiliopoulos, 2018; Yu et al., 2018; Zang et al., 2020), we position our work with respect to three three major classes. One prominent class is the neural-network-as-PDE-solution approach (Bruna et al., 2022; Raissi et al., 2019) which requires the PDE to be known and solves a single system on a single domain. Other methods do not learn the solution directly, but instead augment a PDE-solver as learned corrections (Dresdner et al., 2023; Rackauckas et al., 2021; Um et al., 2021), learned closures (Duraisamy et al., 2019; Sirignano & MacArt, 2023), or learned algorithmic components (Bar & Sochen, 2019; Kochkov et al., 2021). A broader, but less physically constrained approach, is learning a solution operator from the data without knowledge of the governing equations (Cao, 2021; Kovachki et al., 2023; Li et al., 2020, 2021; Lu et al., 2019). While these methods are often evaluated using PDE-generated benchmarks, these are designed to learn directly from data rather than learning to solve a PDE. Neural operators typically do not reach the accuracy of numerical PDE solvers, but they are applicable for domains without explicitly provided equations. This last family is the most similar to our approach, especially Cao (2021) as we use a transformer-based architecture. However, our pretraining procedure is developed for training across multiple operators.

The high cost of training scientific models from scratch has led to significant exploration of transfer learning. Prior work has explored transfer learning in operator networks in such scenarios as conditional shift (Goswami et al., 2022) or new domains, boundary conditions, or distributions over parameters (Li et al., 2021; Subel et al., 2023; Wang et al., 2022a; Xu et al., 2023). However, these too need to be retrained from scratch for new differential operators in the PDE. More recently, efforts have been made to explore transfer across operators and benefits from training on multiple physical systems simultaneously. Subramanian et al. (2023) explores how transfer scales in this setting. However, their study is limited to steady-state linear systems with periodic boundary conditions.

Other works have explored similarly restricted classes or low dimensional, low resolution systems (Desai et al., 2022; Yang et al., 2023).

## 4 Scalable Multiple Physics Pretraining

### 4.1 Compositionality and Pretraining

Many specialized PDEs demonstrate a form of compositionality, as a range of physical phenomena can be described by core components like nonlinear advection or diffusion, but then are augmented or restricted by specialized terms representing concepts like buoyancy or system constraints. To motivate a useful pretraining procedure from this compositionality, we examine two hypotheses:

1. Single models can learn the dynamics for multiple classes of physical behavior.
2. Learning partially overlapping physics is beneficial for transfer learning.

Since many real-world systems share core components, under these hypotheses, training single models on many distinct systems is a natural approach for developing foundation models for physical dynamics. We therefore start by eliminating the complexity related to hypothesis (1) in order to isolate hypothesis (2). We do this by choosing a simple problem setting with one shared scalar field along with consistent scales and geometry: constant-coefficient linear advection-diffusion on a periodic 1D domain. Let $\psi(x, t)$ be a scalar-valued function defined on a periodic spatial domain, $v$ a constant one-dimensional velocity coefficient and $\delta$ a constant diffusion coefficient, then:

$$\textbf{Advection:} \qquad \frac{\partial \psi}{\partial t} + \nabla \cdot (v\psi) = 0$$

$$\textbf{Diffusion:} \qquad \frac{\partial \psi}{\partial t} + \nabla \cdot (-\delta \nabla \psi) = 0$$

$$\textbf{Advection-Diffusion:} \qquad \frac{\partial \psi}{\partial t} + \nabla \cdot (v\psi - \delta \nabla \psi) = 0.$$

If hypothesis (2) holds, we would expect pretraining on advection and diffusion systems individually could be beneficial for transfer to advection-diffusion systems.

We find that this is indeed the case. We pretrain a spatiotemporal transformer model on a large amount of trajectories (100,000 each) with uniformly sampled coefficients ($v \in [-3, -.1] \cup [.1, 3]$, $\delta \in [10^{-3}, 1.]$) generated from the advection and diffusion equations while finetuning on restricted samples from advection-diffusion simulations. The pretrained model is able to achieve much lower error with far fewer samples (Figure 1) without observing advection and diffusion together in the same trajectory during pretraining.

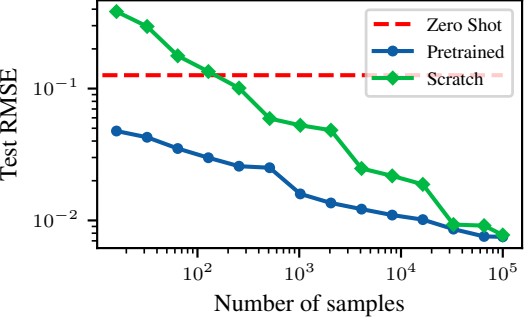

However, to validate hypothesis (1), we must handle much larger spatial resolutions, varying scales, and heterogeneous relationships between fields. Over the rest of this section, we develop an approach for handling these challenges.

Figure 1: Finetuning a model pretrained on large amounts of advection and diffusion data outperforms models trained from scratch on advection-diffusion data across a wide range of data availability (16-100K examples).

### 4.2 Architecture

**Axial Attention.** Given the success of large transformer models in other domains, we employ a scalable axial attention (Dong et al., 2022; Ho et al., 2019; Huang et al., 2019) transformer backbone. For a (2+1)-dimensional system with $T \times H \times W$ tokens, conventional dense attention attends over all tokens simultaneously and has cost $O((HWT)^2)$. Axial attention instead performs a series of attention operations over each axis in turn, limiting the cost to $O(H^2 + W^2 + T^2)$. In Figure 2, it can be seen that while we perform attention on each axis independently, the spatial $K$, $Q$, $V$ projections are shared between the height (y) and width (x) axes.

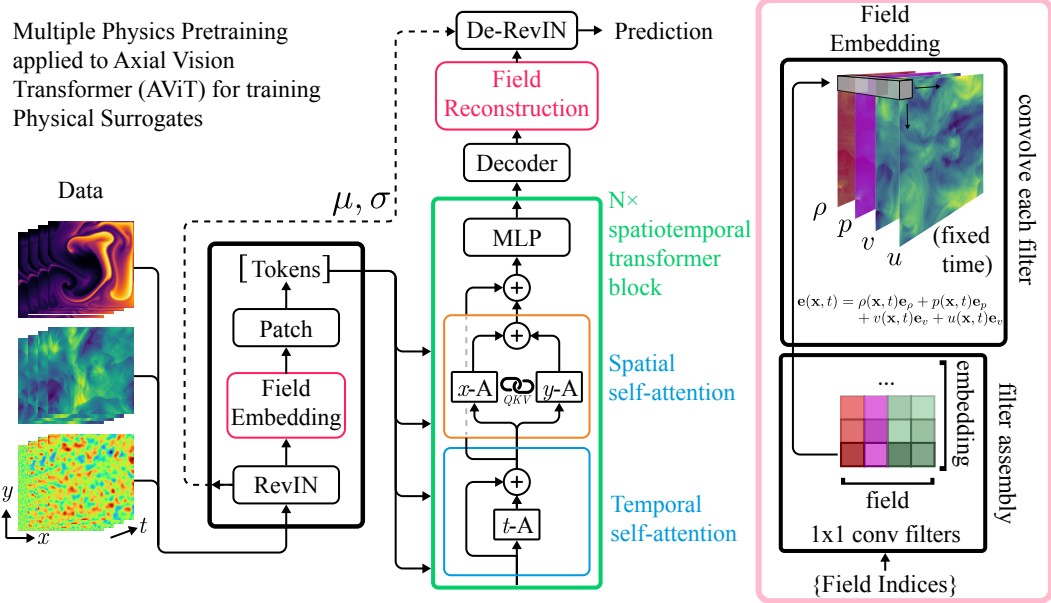

Figure 2: (Left) MPP works by individually normalizing each example using Reversible Instance Normalization (RevIN) then embedding each field individually into a shared, normalized space. A single transformer backbone can then predict the next step for multiple sets of physics. We use an AViT backbone which attends over space and time axis sequentially. Spatial attention is further split by axis, though these share linear projection weights. (Right) The embedding and reconstruction matrices are formed by subsampling a larger $1 \times 1$ convolutional filter based on input fields.

Axial attention has been used in video transformers (Arnab et al., 2021; Bertasius et al., 2021) due to the improved scalability in higher dimensions. While the tools used in our transformer backbone were introduced in prior work, our choice of using fully axial attention differs from ViViT which opted to only separate space and time attention. We favor scalability over maximizing accuracy and so chose fully axial attention. In the following, we refer to this architecture as an Axial ViT (AViT).

**Field Embedding and Normalization.** Embedding multiple physical systems into a single shared representation is complicated by the fact that fields from different systems may operate on entirely different scales in terms of both magnitude and resolution. This is one of the primary challenges that must be addressed for multiple-physics pretraining.

To unify magnitudes, we use Reversible Instance Normalization (Kim et al., 2022, RevIN). We compute the mean and standard deviation of each channel over space-time dimensions and use them to normalize input fields. These statistics are saved and used to denormalize model outputs. While this approach was initially developed for time-series forecasting, in practice the effect is similar to that of the norm scaling utilized in Subramanian et al. (2023).

After rescaling, the data is projected into a shared embedding space. This is the only component with unique weights for each source system. Given a system $S$ with state variables $u(x,t)$, $v(x,t)$, $p(x,t) \in \mathbb{R}$, we project each point or "pixel" into a space of dimension $D^{\text{emb}}$:

$$\boldsymbol{e}(x,t) = u(x,t)\boldsymbol{e}_u + v(x,t)\boldsymbol{e}_v + p(x,t)\boldsymbol{e}_p \tag{1}$$

where $\boldsymbol{e}$ are embedding vectors in $\mathbb{R}^{D^{\text{emb}}}$. This can be seen as a convolution with $1 \times 1$ filters where the input channels of the filter are sub-selected to correspond to the fields present within a given dataset. On the right side of Figure 2, the filter is assembled by sub-selected columns of the larger filter corresponding to the provided fields. It is important to note that this initial projection setup is amenable to fine-tuning to unseen field types. This can be achieved by adding new channels to the initial embeddings, and training them from random initialization. In our models, the shared full resolution space is converted into patched tokens by a sequence of strided convolutions separated by pointwise nonlinearities as in Touvron et al. (2022).

The predictions are reconstructed from the output tokens by reversing this process. The tokens are decoded by a sequence of transposed convolution blocks and projected onto the output fields by taking coordinate-wise inner products with reconstruction vectors $\boldsymbol{r}$:

$$u(x, t + \Delta t) = \langle \boldsymbol{e}(x, t + \Delta t), \boldsymbol{r}_u \rangle. \tag{2}$$

This can similarly be implemented as a $1 \times 1$ convolution with the *output* channels of the convolution filter sub-selected. The mean and standard deviation computed from the inputs are then applied to these normalized outputs to produce the final de-normalized predictions as in Kim et al. (2022).

**Position Biases and Boundaries.** While in most cases, we would like the model to infer boundary conditions from the provided history, we make an exception to this policy for periodic boundaries as they change the continuity of the domain. Transformers are inherently permutation equivariant, and it is essential to include position biases so that the model can learn locality.

With a slight modification, we can use our position biases to capture the change in locality imposed by periodic boundaries. T5-style (Raffel et al., 2020) relative position encodings (RPE) utilize a lookup table to access learned embeddings corresponding to ranges of "relative distance". For periodic boundary conditions, we modify the relative distance computation to account for neighbors across the periodic boundary. In Appendix D.1, we examine simple systems that differ only in boundary conditions and find that this minor change improves generalization in the case where we must learn both periodic and non-periodic conditions.

### 4.3   Balancing Objectives During Training

**Task Sampling.** Our pretraining procedure operates on multiple levels of sampling. The task distribution varies in system $S$, spatial resolution $N_S$, and time resolution $T_S$ and we want diverse batches that accurately capture the signal this provides. However, sampling a full batch from multiple systems at different resolutions simultaneously would be inefficient on modern hardware as it would require batch processing of differently shaped tensors. Multi-GPU training adds an additional complication as the variance in execution time due to unbalanced workloads can lead to inefficient hardware usage.

We mitigate both of these concerns with a simple randomization scheme involving gradient accumulation. Gradient accumulation utilizes multiple backward passes per synchronization step. We therefore sample a single system $S$ uniformly from $\mathcal{S}$ for each *micro-batch*. With $m$ micro-batches per synchronization step, we reduce the work-per-GPU variance $\sigma_{\mathcal{B}}^2$ to $\frac{1}{m}\sigma_{\mathcal{B}}^2$, significantly reducing the average lost cycles due to work discrepancies. This could likely be further reduced by an approximate packing problem solution (Cormen et al., 2022), but we found the random approach was sufficient for our needs. As we employ gradient accumulation in order to increase our batch sizes, this sampling procedure incurs no additional cost.

**Scaled Training Objective.** The simplest approach to obtaining updates from the different tasks is to add their gradients. However, as the magnitudes of the state variables can vary significantly between systems, unweighted losses will result in the gradients from the problems with the largest scales drowning out losses on smaller scales (Yu et al., 2020). To partially control this behavior, we train using the normalized MSE (NMSE) defined as:

$$\mathcal{L}_{\text{NMSE}} = \frac{1}{|\mathcal{B}|} \sum_{S \in \mathcal{S}} \frac{\|\mathcal{M}(\boldsymbol{U}_t^S) - \boldsymbol{u}_{t+1}^S\|_2^2}{\|\boldsymbol{u}_{t+1}^S\|_2^2 + \epsilon} \tag{3}$$

where $\mathcal{B} \subset \mathcal{S}$ denotes the micro-batch and $\epsilon$ is a small number added for numerical stability. This does not account for the full variation in difficulty. Even if sub-task losses have similar magnitudes at the start of training, it is possible for some systems to converge quickly while other losses remain high. Nonetheless, we found that this allows our training process to produce strong results on multiple systems simultaneously.

## 5   Experiments

We design our experiments to probe three vital questions about the utility of MPP:

1. Can large transformer models learn the dynamics of multiple physical systems simultaneously?

Table 1: NRMSE comparison between MPP-pretrained models and dedicated baselines on shallow water equations (SWE), 2D Diffusion-Reaction (DiffRe2D), and compressible Navier-Stokes (CNS) at Mach numbers $M = .1$ and $M = 1$. Complex parameters counted as two real. Top performing within size range and overall are bolded. Dashes indicate precision not provided by source.

| MODEL | #PARAM | SWE | DIFFRE2D | CNS M1.0 | CNS M0.1 |
|---|---|---|---|---|---|
| MPP-AViT-Ti | 7.6M | 0.0066 | **0.0168** | **0.0442** | **0.0312** |
| UNET | 7.7M | 0.083- | 0.84– | 0.4725 | 1.6650 |
| FNO | 927K | **0.0044** | 0.12– | 0.1685 | 0.2425 |
| FNO-B | 115M | 0.00246 | 0.0599 | 0.1451 | 0.1978 |
| ORCA-SWIN-B | 88M | 0.00600 | 0.82– | — | — |
| AViT-B | | | | | |
|   TASK-SPECIFIC | 116M | 0.00047 | 0.0110 | 0.0316 | 0.0261 |
|   MPP | 116M | 0.00240 | 0.0106 | 0.0281 | 0.0172 |
|   MPP + FINETUNED | 116M | **0.00043** | **0.0087** | **0.0187** | **0.0079** |
| MPP-AViT-S | 29M | 0.0039 | 0.0112 | 0.0319 | 0.0213 |
| MPP-AViT-L | 409M | 0.0022 | 0.0098 | 0.0208 | 0.0147 |

2. Does pretraining lead to improved accuracy on previously unseen physics?

3. Does MPP provide a finetuning advantage over existing spatiotemporal foundation models?

**Data.** We use the full collection of two-dimensional time-dependent simulations from PDEBench (Takamoto et al., 2022) as our primary source for diverse pretraining data. This includes systems governed by four unique nonlinear PDEs defined over a variety of state variables, resolutions, initial conditions, boundary conditions, and simulation parameters. The specific PDEs are the compressible and incompressible Navier-Stokes equations (CNS/INS), the shallow-water equations (SWE), and a 2D Diffusion-Reaction equation (DiffRe2D). Full details on the data used can be found in Appendix B.1.

**Training settings.** $T_S$ is fixed at 16 for all experiments as our VideoMAE comparison in Section 5.2 was unable to scale to larger sizes without gradient checkpointing. Autoregressive training is performed only one step ahead—no longer rollouts, noise corruption, or post-processing are included for stability. Training from scratch and MPP pretraining are always performed on the AViT architecture described in section 4.2. Full training details including data splits, optimization details, and hardware are documented in Appendix C.

### 5.1 Pretraining Representations

First, we evaluate whether pretraining on multiple task actually leads to effective representations by comparing MPP-pretrained models to dedicated baselines from prior work across all available systems. The models are pretrained at a variety of sizes so we can begin to explore to benefits of scaling our approach. Precise model sizes can be found in Appendix C.1. Unlike the baselines which are trained on only one system and so must only learn one parameter regime, our models (denoted by MPP-AViT-*) must handle all systems and regimes without finetuning. The effect of physical parameters, forcing, and simulation parameters must be inferred from context $U_t^S$. The UNet (Ronneberger et al., 2015) and FNO (Li et al., 2020) results are sourced from Takamoto et al. (2022) while the results from Shen et al. (2023) with a finetuned SWIN (Liu et al., 2021) are used for ORCA. As the lightweight FNO proved to be the most competitive comparison, we train an additional FNO beyond the PDEBench results that has been scaled to 115M parameters (labeled "FNO-B") for fairness. Results are reported in terms of Normalized RMSE (NRMSE, the square root of Equation 3) averaged over fields and examples, as in Takamoto et al. (2023). Our Compressible Navier-Stokes results are aggregated based on the mach number here due to space limitations. Fully granular results can be found in Appendix D.4. Ablation results demonstrating the importance of balanced losses and normalization are shown in Appendix D.2.

Our pretrained models are able achieve high-end performance on all datasets (Table 1) despite the difficulty of multi-task training (Yu et al., 2020) while showing improved performance with scale. Our smallest pretrained model, the MPP-AViT-Ti outperforms the PDEBench baselines on all problems except for SWE. However, though both models improve in absolute performance with scale, the

pretrained AViT-B catches up to the FNO-B. It is important to clarify that we are not claiming the pretrained models are optimal—with a series of comparisons on the AViT-B models, we show that at times, the multi-task training does hurt performance on individual tasks and that we can improve upon the pretrained model performance by finetuning our own models on specific tasks. What this experiment answers affirmatively is that large transformers can learn multiple sets of dynamics simultaneously. Trajectories from pretrained models are displayed in Appendix D.6.

## 5.2 Transfer to Low-data Domains

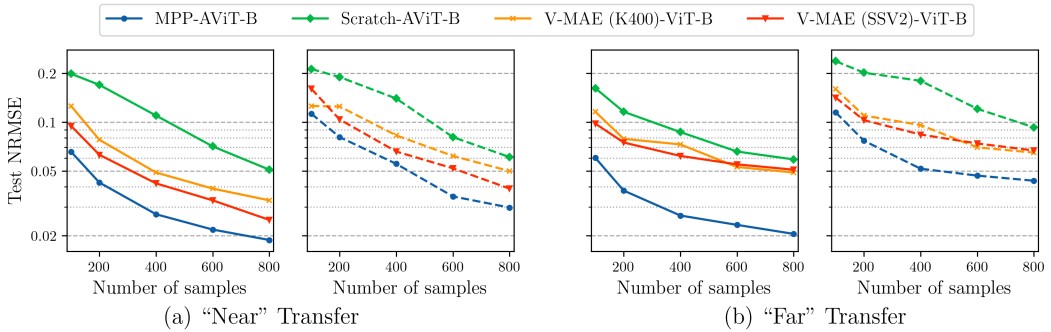

(a) "Near" Transfer  (b) "Far" Transfer

Figure 3: NRMSE for transfer learning tasks. Solid lines are one-step error. Dashed lines are averaged error over five step rollouts. The MPP model shows clear performance benefits in both cases. The more turbulent behavior of "far" seems to be difficult to learn from scratch or from video data, but pretraining on physical data leads to much stronger results.

Harkening back to Section 4.1, since we have now shown that we can learn multiple sets of dynamics with a single model, we return to the question of whether these multiple physics models are well suited to transfer learning. For a more realistic exploration of transfer, we construct a setting where the model must learn new physical behavior by removing all compressible fluid data from the pretraining corpus and pretraining on the three remaining spatiotemporal systems. We then evaluate transfer to two specific compressible Navier-Stokes datasets:

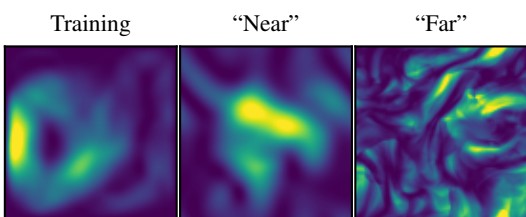

Figure 4: Kinetic energy for incompressible pretraining and compressible finetuning examples. The "near" compressible snapshot resembles the pretraining snapshot while "far" displays new turbulent small scales.

- *"Near":* $M = 0.1$, viscosity$= 10^{-2}$, Random Periodic Initial Conditions

- *"Far":* $M = 1.0$, viscosity$= 10^{-8}$, Turbulent Initial Conditions

Snapshots of the kinetic energy for the finetuning systems and incompressible pretraining data are shown in Figure 4. While quantitatively evaluating the physics gap is an unsolved problem, the names reflect both prior physical knowledge and qualitative evaluation. "Near" features a low Mach number, the dimensionless quantity that correlates with compressible behavior, and viscosity similar to that of the incompressible simulation. "Far" has wildly different turbulent behavior that induces small scale structure never seen during training. However, despite the similarity in physical behavior, the simulations are still quite different: the compressible and incompressible simulations in PDEBench differ in spatial and temporal resolution, initial condition distribution, boundary conditions, viscosity, and velocity range in addition to the difference in compressibility. We use these sets to compare the finetuning performance of MPP, training from scratch, and an existing pretrained spatiotemporal transformer, VideoMAE (Tong et al., 2022) pretrained on both K400 (Kay et al., 2017) and SSV2 (Goyal et al., 2017) datasets. Details on the finetuning procedure followed can be found in Appendix C.3.1.

Figure 3 shows that the MPP models outperform VideoMAE and training from scratch by a large margin in the low-data regime. Numerical results are listed in Appendix C. VideoMAE displays surprisingly strong finetuning performance given that the pretraining data is conventional video, but it is unable to match the much lower memory (VideoMAE at 79.3 GB vs. AViT-B at 24.7 GB peak VRAM for batch size 1) MPP-AViT-B in either setting. Predictably, both pretraining approaches are less accurate in the long-run on the turbulent "far" dataset. However, in the short-term the physical pretraining seems to provide an even larger advantage in this regime compared to the far smoother "near" data. Rollout visualizations are included in Appendix D.7.

One possible explanation for the strong performance of finetuning for "far" is that this experiment can be viewed as a more realistic example of the compositionality exploration in Section 4.1 from the perspective of classification of second-order PDEs. The solution to Navier-Stokes in the vanishing viscosity limit represents one possible weak solution to the Euler equations which are classically hyperbolic (LeVeque & Leveque, 1992). While the viscous incompressible flow from pretraining is governed by the same transport equations as "far", those solutions are dominated by smoothing locally parabolic behavior. However, the inviscid shallow water simulations in the training set are hyperbolic. The model has therefore seen two major components of "far", but has never seen them within one system until finetuning.

## 5.3 Inflation to 3D

While 2D problems offer a compelling middle ground between complexity and cost for experimentation, most physical phenomena of real-world interest are fundamentally three-dimensional. We therefore examine the usefulness of our pretrained models when "inflated" to 3D. Inflation techniques were first demonstrated in Carreira & Zisserman (2017) and have seen use for extending 2D visual (image) classifiers to 3D spatiotemporal (video) settings (Nguyen et al., 2022; Xie et al., 2018). Here we employ the technique to add an additional spatial dimension.

The factored architecture of the AViT is well-suited to inflation. We initialize the projection weights discussed in Section 4.2 to those from the 2D compressible Navier-Stokes data seen during training with the new velocity direction initialized as the average of the previous two velocity projections. Since the transformer backbone acts on each spatial axis independently, the only dimension-dependent operations are the learned downsampling or "patching" convolutions. These convolutional layers are modified by following the inflation procedure of Carreira & Zisserman (2017): a 2D kernel of size $P \times P$ is inflated into a 3D kernel by repeating the 2D kernel $P$ times along the new axis and rescaling by $\frac{1}{P}$. This gives us a 3D convolutional operation that is equivalent to applying previous 2D filters then average pooling in the new direction. In practice, we find performance can be slightly improved by adding low magnitude Gaussian noise to the resulting filter. Due to the low resolution of the "Turbulent" dataset, we additionally reduced the stride of the first convolution in the hMLP by a factor of 2 for a total downsampling factor of 8 rather than the 16 used elsewhere. Standard training details are found in Appendix C.4

We compare these inflated 2D to 3D models to both training an identical architecture from scratch and PDEBench baselines. Unlike in Section 5.2 where we held out the 2D Compressible Navier-Stokes data, we use the full pretrained models from section 5.1 here. Nonetheless, this remains a significant physical gap as 2D and 3D turbulence are well-understood to have major differences in behavior (Ouellette, 2012). Due to the difficulty of scaling 3D training, results in this section are reported at the "Ti" scale.

Table 2: NRMSE for 2D to 3D inflation. Sub-headings are initial condition type.

| SIZE (B, T, N) | TURBULENT $(600, 21, 64^3)$ | RANDOM $(100, 21, 128^3)$ |
|---|---|---|
| FNO | 0.240 | 0.370 |
| UNET | 0.230 | 1.000 |
| AViT-SCRATCH | 0.098 | 0.299 |
| AViT-MPP | **0.094** | **0.264** |

Table 2 demonstrates significant improvements on both the turbulently and randomly initialized Compressible Navier-Stokes datasets from PDEBench with a $11.7\%$ improvement for the smaller dataset of randomly initialized simulations and even a $4.1\%$ improvement for the larger turbulently initialized dataset where both the dimensionality and sampling are adjusted. Training on high resolution 3D data is an enormously expensive procedure. Our results suggest that pretraining on 2D data and inflating to 3D is a promising strategy for developing models that can be used in this space.

# 6  Conclusion

We introduced an autoregressive pretraining strategy, Multiple Physics Pretraining, for the development of multi-use physical surrogates. Through per-sample normalization, field embeddings, appropriately scaled losses, and efficient task sampling, we are able to train scalable transformer models capable of predicting multiple sets of independent dynamics simultaneously. We evaluated several sizes of model and observed that the approach benefits from scale. MPP models were able to match dedicated modern baselines on benchmarks containing fluid and reaction simulations derived from multiple equations, simulation parameters, and boundary conditions from pretraining alone with even stronger performance after undergoing task-specific finetuning. Our pretrained models also showed positive transfer by outperforming both training from scratch and existing video models on previously unseen physics. Furthermore, through kernel inflation approaches, we were able to demonstrate improved results on 3D simulation compared to training from scratch.

**Limitations and Future Work.** The focus of our work is on transfer and learning from multiple data sources, however, many interesting questions remain before we can develop true foundation models for spatiotemporal physics. For example, our choice of architecture enabled us to scale to some of the higher resolution benchmarks available today, but it also limits our models to uniform grids. Extensions to the non-uniform geometries frequently encountered in physical simulation will require further work balancing efficiency and flexibility. Extending to 3D, for example, is an enormously expensive task for many families of models. Our inflation approach is promising and future work can help identify better approaches for performing this inflation. In terms of more physically motivated concerns, we must also explore the role of constraints and conservation laws in models developed for multiple physical systems. Hard constraints that are necessary for a closed system may ensure error in an open system, thus adaptable approaches may need to be developed.

Similarly, many physically interesting tasks require handling incomplete or noisy data. While certain datasets in PDEBench do not provide all relevant physical fields (the Shallow Water equations, for instance, only contain $h$ rather than full velocity fields needed to simulate the data) all the data that is available is observed at every grid point without noise. It remains to be seen if these approaches generalize to settings without this type of clean data.

Finally, as these are early days in the development of foundation models for physics, the limits of transfer in these spaces are poorly understood. PDEBench, which we used here, is constructed from largely fluid-like data. While we demonstrated strong transfer benefits from pretraining, it remains to be seen how far away from the training distribution or training tasks these benefits persist. It is even an open question how to define distance in this space. Future work will need to expand data diversity to push these questions and others forward. MPP opens up many new research directions and paves the way for this development in the future.

## Acknowledgments

The computations in this work were, in part, run at facilities supported by the Scientific Computing Core at the Flatiron Institute, a division of the Simons Foundation. M.P. is supported by the Department of Energy, Office of Science under contract number DE-AC02-05CH11231. M.M. would like to thank Jed Brown for his valuable insight on the simulation data. Polymathic AI acknowledges support provided by the Simons Foundation and Schmidt Sciences, LLC.

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

# A Impact Statement

This paper contributes to the field of machine learning for computational physics. In terms of positive impact, we would hope that our approach enables researchers and engineers to develop more accurate dynamics models from limited observations. While we see mostly positive implications from this, the danger of any computational physics research is potential use in weapons research. In this case, we feel that risk is fairly small as research in such spaces tends to focus on well-understood physics where numerical methods already obtain extremely high precision while our approach is more suited to the more scientifically oriented space of modeling poorly understood physics at lower precision.

# B Data Details

## B.1 PDEBench

To train and evaluate our models, we use the publicly available PDEBench dataset[2] (Takamoto et al., 2022). We summarize the data included in this section. This dataset comprises a suite of time dependent and time independent simulations based on common PDE systems, generated with varying parameters, initial conditions, and boundary conditions. Specifically, PDEBench uses a discretized ground-truth solver with high precision to evolve the vector-valued solution to a given PDE at one time step to the solution at one time step later. When compiled across time steps, the vector-valued solutions take the form $x \in \mathbb{R}^{T \times C \times H \times W}$, where $T$ denotes the total number of times steps, $H$ and $W$ denote the spatial height and width of the simulation grid and $C$ denotes the parameter space representing the velocity ($v_x$ and $v_y$), pressure ($p$) and density ($\rho$) fields, such that $C = 4$. For our study, we focus on the 2D fluid dynamics simulations in PDEBench. These are outlined loosely below; for more details, we refer the reader to Takamoto et al. (2022):

**Compressible Navier-Stokes:** These equations are used to model the pressure and velocity of both laminar and turbulent Newtonian fluids, and are applied to many real-world problems, from aerodynamics to interstellar gas dynamics. In the regime in which the density of the fluid can change due to pressure variation, the equations can be expressed:

$$\partial_t \rho + \nabla \cdot (\rho \mathbf{v}) = 0, \tag{4}$$

$$\rho \left( \partial_t \mathbf{v} + \mathbf{v} \cdot \nabla \mathbf{v} \right) = -\nabla p + \eta \nabla^2 \mathbf{v} + (\zeta + \eta/3) \nabla (\nabla \cdot \mathbf{v}) \tag{5}$$

$$\partial_t (\epsilon + \rho v^2 / 2) + \nabla \cdot \left[ (p + \epsilon + p v^2 / 2) \mathbf{v} - \mathbf{v} \cdot \boldsymbol{\sigma}' \right] = \mathbf{0}, \tag{6}$$

where $\rho$ is the fluid density, $\mathbf{v}$ is the fluid velocity, $p$ is the fluid pressure, $\epsilon$ is the internal energy, $\boldsymbol{\sigma}'$ is the viscous stress tensor, $\eta$ is the shear viscosity, and $\zeta$ is the bulk viscosity. For our transfer experiments, we use the following two sets of data in particular:

1. A set of 1,000 trajectories on a $H \times W = 512 \times 512$ regular grid over $T = 100$ time steps (where the separation between steps is $\Delta t = 0.005$). Additionally, $(M, \eta, \zeta) = (1.0, 10^{-8}, 10^{-8})$, where $M$, $\eta$, $\zeta$ denote the Mach number, the shear viscosity, and the bulk viscosity, respectively. The velocity field is initialized with a turbulent field, while the inital pressure and density fields are taken to be uniform.

2. A set of 10,000 trajectories on a $H \times W = 128 \times 128$ regular grid with $(M, \eta, \zeta) = (0.1, 0.01, 0.01)$. The time steps and initializations are as above.

**Incompressible NS:** In the incompressible regime, which typically occurs in fluids with low Mach numbers (as it rules out density and pressure waves like sound or shock waves), the Navier-Stokes equations simplify to:

$$\nabla \cdot \mathbf{v} = 0, \tag{7}$$

$$\rho \left( \partial_t \mathbf{v} + \mathbf{v} \cdot \nabla \mathbf{v} \right) = -\nabla p + \eta \nabla^2 \mathbf{v} + \mathbf{f}, \tag{8}$$

where $\mathbf{v}$ is the velocity, $\rho$ is the density, $p$ is the pressure, $\eta$ is the viscosity, and $\mathbf{f}$ is a spatially varying external force. The simulation in PDE bench is augmented by an immersed tracer that is transported by the velocity field:

$$\partial_t \rho_{smoke} = -\mathbf{v} \cdot \nabla \rho_{smoke} \tag{9}$$

---

[2]https://github.com/pdebench/PDEBench

The system uses Dirichlet boundary conditions on the velocity field $\mathbf{v} = 0$ and Neumann on the density $\frac{\partial \rho_{smoke}}{\partial x} = 0$, $x \in \{0, 1\}$, $\frac{\partial \rho_{smoke}}{\partial y} = 0$, $y \in \{0, 1\}$,. These equations are typically used to model a variety of hydrodynamics systems such as weather. This data is produced at resolution $512 \times 512$ with time step of .0005. The dataset contains a total of 1000 trajectories with 1000 time steps each.

**Shallow water:** In the event that the horizontal length scale of the fluid is significantly greater than the vertical length scale, the incompressible Navier-Stokes equations can be depth-integrated to derive the shallow water equations. These describe flow below a pressure surface in a fluid, and are given by

$$\partial_t h + \nabla \cdot (h\mathbf{v}) = 0, \tag{10}$$

$$\partial_t(h\mathbf{v}) + \nabla \cdot \left(\frac{1}{2}h\mathbf{v}^2 + \frac{1}{2}g_r h^2\right) = -g_r h\nabla b, \tag{11}$$

where $h$ is the water depth, $\mathbf{v}$ is the velocity, $b$ is the bathymetry, and $g_r$ is the reduced gravity. For our data, we use 1,000 trajectories on a $H \times W = 128 \times 128$ regular grid over $T = 100$ time steps. The specific simulation used is a 2D radial dam break scenario, where the water height is initialized as a circular bump in the center of the domain with a uniformly randomly sampled radius.

**Diffusion-Reaction:** The Diffusion-Reaction equations arise in systems with many interacting components and can be represented in the general form

$$\partial_t \mathbf{u} = \mathbf{D}\nabla^2 \mathbf{u} + \mathbf{R}(\mathbf{u}), \tag{12}$$

where $\mathbf{u}$ is a vector of concentration variables, $\mathbf{D}$ is a diagonal matrix of diffusion coefficients, and $\mathbf{R}$ describes all local reaction kinetics. The most common application of diffusion-reaction equations is in chemical reactions, however they can also be used to describe a variety of dynamical processes. For our data, we use 1,000 trajectories on a $H \times W = 128 \times 128$ regular grid over $T = 100$ time steps. The reaction functions for the activator and inhibitor are defined by the Fitzhugh-Nagumo equation (Klaasen & Troy, 1984), and their diffusion coefficients are $D_u = 1 \times 10^{-3}$ and $D_v = 5 \times 10^{-3}$ respectively. The initial conditions are generated as standard Gaussian random noise.

### B.2 PDEArena

In addition to the 2D Incompressible Navier-Stokes data incorporated from PDEBench, we also include 2D Incompressible Navier-Stokes data from PDEArena (Gupta & Brandstetter, 2022).

These follow roughly Equation 7, with a minor variation:

$$\nabla \cdot \mathbf{v} = 0, \tag{13}$$

$$\rho\left(\partial_t \mathbf{v} + \mathbf{v} \cdot \nabla \mathbf{v}\right) = -\nabla p + \eta \nabla^2 \mathbf{v} + \begin{bmatrix} b \\ 0 \end{bmatrix}, \tag{14}$$

where $b \in [0.2, 0.5]$ represents buoyancy in the $y$ direction. Unlike PDEBench, this is a spatially constant term. This includes a set of 5,200 training trajectories (and 1,300 validation and test trajectories each) on a $H \times W = 128 \times 128$ regular grid from which we take $T = 16$ timesteps for prediction. As with the PDEBench simulations, the PDEArena simulations include a viscosity parameters of $\nu = 0.01$ and Dirichlet boundary conditions.

## C  Experiment Details

### C.1  Model Configurations

The following architectural decisions were used across all AViT models trained in this paper:

- **Pre/Post Norm:** Pre-norm (Xiong et al., 2020)
- **Normalization Type:** Instance Normalization (Ulyanov et al., 2017)
- **Activations:** GeLU (Hendrycks & Gimpel, 2016)
- **QK Norm:** Yes (Dehghani et al., 2023)
- **Patching:** hMLP (Touvron et al., 2022)

Table 3: Details of the various model architectures and scales explored.

| Model | Embed Dim. | MLP Dim. | # Heads | # Blocks | Patch Size | # Params |
|-------|-----------|----------|---------|----------|------------|----------|
| AViT-Ti | 192 | 768 | 3 | 12 | [16, 16] | 7.6M |
| AViT-S | 384 | 1536 | 6 | 12 | [16, 16] | 29M |
| AViT-B | 768 | 3072 | 12 | 12 | [16, 16] | 116M |
| AViT-L | 1024 | 4096 | 16 | 24 | [16, 16] | 409M |

Table 4: Inference time for various models on A6000 GPU.

| Model | Time (ms) |
|-------|-----------|
| UNet | 67.7 |
| FNO | 7.2 |
| AViT-Ti | 19.2 |
| ORCA-SWIN-B | 98.5 |
| AViT-B | 105.6 |

- **Decoder:** Transposed hMLP (this is equivalent to the transposed convolutions mentioned in the main text).
- **Causal Masking:** False - We only evaluate the loss on the $T + 1$ prediction.

Furthermore, we examine the performance of our models on the aforementioned PDE systems when the size of the model is scaled. Vision transformers have a variety of parameters that control the model's size, including the number of processor blocks, the dimensionality of patch embeddings and self-attention, the dimensionality of Multi-Layer Perceptron (MLP) blocks, the number of attention heads, and the patch size applied on the input tensors. In previous studies on language (Hernandez et al., 2021; Hoffmann et al., 2022; Kaplan et al., 2020) and vision (Zhai et al., 2022), it has generally been noted that model performance is typically only weakly dependent on shape parameters, and instead depends largely on non-embedding parameter count given a fixed compute budget and dataset size. As such, we follow the general scaled architectures set forth by Zhai et al. (2022) for vision, and scale all aspects of the model shapes simultaneously to select a variety of model sizes for testing. These are detailed in 3. In practice, the AViT models are on average slightly slower than the similarly sized baseline models from Table 1.

**Software.** All model development and training in this paper is performed using PyTorch 2.0 (Paszke et al., 2019).

**Hardware.** All training for both pretraining and finetuning is done using Distributed Data Parallel (DDP) across 8 Nvidia H100-80GB GPUs.

### C.2 Exp 1: Pretraining Performance

For MPP, we train using the following settings:

- **Training Duration:** 200K steps
- **Train/Val/Test:** .8/.1/.1 split per dataset on the trajectory level.
- **Task sampling:** Uniformly sample task, then uniformly sample trajectory from task without replacement. We treat every 400 model updates (1 model update=5 micro-batches) as an "epoch" and reset the task pool.
- **Micro-batch size:** 8
- **Accumulation Steps:** 5
- **Optimizer:** Adan (Xie et al., 2023)
- **Weight Decay:** 1E-3
- **Drop Path:** 0.1

- **Base LR:** DAdaptation (Defazio & Mishchenko, 2023)
- **LR Schedule:** Cosine decay
- **Gradient clipping:** 1.0

For training from scratch, no task sampling occurs and sampling without replacement continues until the dataset is exhausted as in a conventional epoch. Note, we use the automated learning selection strategy DAdaptation during pretraining runs in large part to avoid excessive hyperparameter tuning of our own models. In finetuning experiments, comparison models are tuned manually following the recommended settings from the model publishers to avoid differences being due to compatibility with the parameter-free method.

**FNO-B** For the scaled-up FNO, we modify the parameters used in PDEBench to 6 layers, 24 modes, and width 100. Training configurations were taken from PDEBench (500 full passes through the dataset with the Adam optimizer) with learning rate selected by DAdaptation as in the training of our models for consistency.

**Data** For pretraining, we use all 2D time-dependent PDEBench datasets. These are described in Section B.1. In particular, we use the compressible and incompressible Navier-Stokes, Diffusion-Reaction 2D, and Shallow Water data.

## C.3 Experiment 2: Transfer to Low-Data Domains

In this experiment, we compare the transferability of our MPP-Pretrained models to general-purposes pretrained video masked autoencoders (VideoMAE; Tong et al., 2022) for frame prediction on video-like PDEBench data (Takamoto et al., 2022).

**Data** We study transferability of VideoMAE models for spatiotemporal prediction on video-like scientific data.

MPP-labeled models are pretrained on datasets generated from three PDEs: Incompressible Navier-Stokes, Shallow Water, and Diffusion Reaction 2D. This is performed using the same training settings as in Section C.2.

We focus on transfer to the two datasets *"Near"* and *"Far"* (see Sect. 5.2) of fluid dynamics simulations taken from the PDEBench dataset (Takamoto et al., 2022). These simulations solve the compressible Navier-Stokes equations in a 2D geometry with periodic boundary conditions (see Appendix B.1 for additional details).

### C.3.1 MPP Finetuning Procedure

For MPP and training from scatch, we use the following settings:

- **Training Duration:** 500 epochs (true epochs due to the restricted dataset)
- **Train/Val/Test:** X/.1/.1 split per dataset on the trajectory level. Note that X is due to the fact that we test varying amounts of training data. These are subsampled from the training split of 80%.
- **Batch size:** 8
- **Accumulation Steps:** 1 (No accumulation)
- **Optimizer:** Adan (Xie et al., 2023)
- **Weight Decay:** 1E-3
- **Drop Path:** 0.1
- **Base LR:** DAdaptation (Defazio & Mishchenko, 2023)
- **LR Schedule:** Cosine decay
- **Gradient clipping:** 1.0

**Channel expansion.** The only component of the architecture used that is aware of the particular fields being ingested is the field embedding and debedding projection layers. As the finetuning data adds previously unseen state variables, this matrix needs to be expanded. To be concrete, out of the four state variables present in the CNS data – $v_x$, $v_y$, $P$, $\rho$ – the training data only contains $v_x$, $v_y$, $\rho$. If the original field embedding layer was therefore a $1x1$ convolution with 3 input channels, we follow the following procedure to add the new projection.

1. Instantiate a new `field_projection` and `field_reconstruction` layer with a sufficiently large number of channels to process both previously seen fields and new fields.

2. If $W$ is a $(..., C_{in}, C_{out})$ `field_projection` weight matrix, set $W^{new}[..., : C_{in}^{old}, :] = W^{old}$. Perform the corresponding replacement within the `field_reconstruction` layer.

All other weights are loaded as normal without modification.

**Weights trained.** We found that partially frozen training resulted in noticeably worse performance and therefore finetuned the full model.

### C.3.2 VideoMAE Settings

While VideoMAE does utilize spatiotemporal information, it was developed for a different setting, so we fully document all details of our adaptation of it here both for reproducibility and fairness in our comparison.

VideoMAE models are video transformers that were proven to be efficient data-learners for self-supervised video pretraining (Tong et al., 2022). They rely on an asymmetric encoder-decoder architecture building on a vanilla ViT backbone with joint space-time attention. VideoMAE models are pretrained by learning to reconstruct masked videos using a random tube-masking strategy with a extremely high masking ratio ($\sim 90\,\%$).

We make use of two publicly available models, hereafter called VideoMAE-K400 and VideoMAE-SSV2, that were pretrained on Kinetics-400 dataset (K400; Kay et al., 2017) and Something-Something V2 dataset (SSV2; Goyal et al., 2017), respectively. Both datasets are made of short videos (typically $\leq 10\,s$ long) of human-object or human-human interactions. VideoMAE-K400 (respectively, VideoMAE-SSV2) was pretrained on $\sim 240$k ($\sim 170$k) videos. We focus on the models that build on a ViT-base backbone, so that their size (in terms of number of trainable parameters) remains comparable to that of MPP-AViT-B. After adaptation of the input and output linear layers as described below, the number of trainable parameters of these models reaches $\sim 95$ M.

**Number of channels.** Same as the original pretraining procedure, the input data $x \in \mathbb{R}^{C \times T \times H \times W}$ is divided into non-overlapping joint space-time cubes of size $2 \times 16 \times 16$. These are embedded through a `Conv3d` layer, resulting in $\frac{T}{2} \times \frac{H}{16} \times \frac{W}{16}$ tokens. Since our PDEBench data has $C = 4$ channels instead of 3 for the RGB videos from the pretraining set, we had to adapt the number of input channels of this `Conv3d` layer accordingly. The weights of this new layer were defined using a (rescaled) repetition of the pretrained weights from the original layer. Similarly, the output number of features of the final linear projection layer of the model had to be adapted to $C = 4$ channels. The weights and biases of this layer were extended by consistently repeating the original pretrained weights and biases.

**Positional encoding.** The number of tokens resulting from our PDEBench data did not match the number of tokens resulting from the pretraining datasets. Consequently, we also had to adapt the pretraining positional encoding. We chose to interpolate accordingly the original 1D sine/cosine positional encoding (Vaswani et al., 2017) using a trilinear interpolation after having reshaped the token index axis onto a 3D grid.

### C.3.3 Video MAE Finetuning Procedure

We describe the finetuning procedure of the pretrained VideoMAE models for frame prediction. Frame prediction consists in predicting the next $T_p$ frames of a video given a context of $T_c$ frames. Since the pretrained models manipulates space-time cubes of size 2 in time, we naturally choose $T_p = 2$. The context size is taken to be $T_c = 16$ for consistency with MPP-AViT models. We finetune

Table 5: Effective learning rate for the finetuning of VideoMAE.

|  | "Near" | "Far" |
|---|---|---|
| VIDEOMAE (K400) | 0.00039 | 0.00198 |
| VIDEOMAE (SSV2) | 0.00186 | 0.00150 |

the pretrained models for frame prediction by adapting the self-supervised training strategy in order to reconstruct the last $T_p$ frames of a masked video of $T = T_c + T_p$ frames.

**Masking strategy.** For frame prediction, instead of the random tube-masking strategy, we simply mask the last $T_p$ frames of the input data.

**Loss.** We finetune our models by minimizing a NMSE loss. In this context, denoting by $x, y \in \mathbb{R}^{C \times T_p \times H \times W}$ the output of our model and the target (masked frames), respectively, the NMSE loss is defined by $\mathcal{L}(x, y) = \sum_{c=1}^{C} \sum_{t=1}^{T_p} \|x_{c,t} - y_{c,t}\|_2^2 / \|y_{c,t}\|_2^2$.

**Normalization of the data.** Each set of PDEBench simulations is globally and channel-wise rescaled so that pixel values all fit in $[0, 1]$. Additionally, we normalize channel-wise the targets $y \in \mathbb{R}^{C \times T_p \times H \times W}$ by subtracting the global mean of the corresponding context frames and then dividing by their global standard deviation.

**Optimization.** We finetune the pretrained models over 500 epochs (full epochs due to restricted data size) and a (total) batch size of 8 using AdamW optimizer (Loshchilov & Hutter, 2019). Except for the learning rate, the remaining optimization hyperparameters are chosen to be consistent with those used in the finetuning experiments of Tong et al. (2022) (Table 10). In particular, we choose a weight decay $\lambda = 0.05$, $(\beta_1, \beta_2) = (0.9, 0.999)$, a cosine learning rate decay scheduler with 5 warmup epochs, a drop path rate of 0.1, and a layer-wise learning rate decay parametrized by 0.75. In this setting, the learning rate is adjusted by performing a hyperparameter search monitored with WandB (Biewald, 2020). We report the resulting optimal values per pretrained model and dataset in Table 5.

### C.4 Experiment 3: Inflation to 3D

For MPP and training from scatch, we use the following settings:

- **Training Duration:** 100 True Epochs

- **Train/Val/Test:** .8/.1/.1

- **Batch size:** 8

- **Accumulation Steps:** 1 (No accumulation)

- **Optimizer:** Adan (Xie et al., 2023)

- **Weight Decay:** 1E-3

- **Drop Path:** 0.1

- **Base LR:** DAdaptation (Defazio & Mishchenko, 2023)

- **LR Schedule:** Cosine decay

- **Gradient clipping:** 1.0

**Data** We use both 3D Compressible Navier-Stokes datasets provided by PDEBench for these experiments. The first has 600 total trajectories of 21 steps simulated with sheer and bulk viscosities of $10^{-8}$ with explicitly turbulent initialization. The second contains 100 total trajectories of 21 steps with viscosities of $10^{-8}$ and random initialization.

Table 6: Validation NRMSE for position bias comparison. Compares training performance on data that differs only in boundary conditions.

| TRAINING | PERIODIC | ABSORBING |
|---|---|---|
| PERIODIC BASELINE | 0.032 | — |
| ABSORBING BASELINE | — | 0.295 |
| COMBINED | | |
|    STANDARD RPE | 0.188 | 0.189 |
|    PERIODIC-ADJUSTED RPE | 0.081 | 0.143 |

## C.5 Appendix Experiment: Broader Usage of Pretrained Representations

For MPP and training from scatch, we use the following settings:

- **Training Duration:** 20K Optimization Steps
- **Train/Val/Test:** 1000/100/1000 taken from original validation set or randomly depending on whether data was used for training.
- **Batch size:** 24
- **Accumulation Steps:** 1 (No accumulation)
- **Optimizer:** Adan (Xie et al., 2023)
- **Weight Decay:** 1E-3
- **Drop Path:** 0.1
- **Base LR:** DAdaptation (Defazio & Mishchenko, 2023)
- **LR Schedule:** Cosine decay
- **Gradient clipping:** 1.0

## D   Additional and Extended Results

### D.1   Position Bias Evaluation

We isolate the impact of position biases on our multi-task training objectives by constructing an experiment that isolates their influence. Recall the advection equation from Equation 1:

$$\frac{\partial \psi}{\partial t} + \nabla \cdot (v\psi) = 0 \tag{15}$$

We will define two sets of physics. In both cases, the function is defined on the 1D domain $x \in [0, 1]$. We sample $v \sim Unif(-1, 1)$ and use initial conditions sampled from the set of circular Gaussians with variances sampled from $Unif(1/160, 1/5)$ and means sampled from $Unif(.25, .75)$. The two systems vary only in the choice of boundary conditions. The first uses periodic boundary conditions, implying $\phi(0) = \phi(1)$. The second uses absorbing boundary conditions in which waves are not reflected back into the solution space. The restricted functional form allows us to implement this exactly by extending the domain and solving the periodic equations such that the constant velocity implies the waves exiting the solution space never return.

In this experiment, we first train models (AViT-Ti with 1D patches) on each system individually using 10,000 examples each for 100 epochs to get a sense of the baseline performance. We then train models with and without our modified position biases on the two systems jointly (20,000 examples) to evaluate the impact of our change.

Table 6 shows that our modified position biases are more effective at training in the joint setting. Both RPE schemes are able to improve on absorbing boundary with the additional data. Standard RPE on the other hand struggles to learn the periodic baseline. Our Periodic-adjusted variant is much more effective at learning the periodic data, though it does not outperform the baseline.

It is interesting to note how large the effect of boundary conditions is on this problem. The model trained on only periodic condition reaches nearly an order of magnitude higher precision. While

Table 7: Ablation table showing changes in NRMSE as our proposed modifications to the architecture and model are removed.

| SETTING | SWE | DIFFRE2D | CNS M1.0 | CNS M0.1 |
|---|---|---|---|---|
| MPP-AViT-B | 0.00240 | 0.0106 | 0.0281 | 0.0172 |
| REMOVE NRMSE TRAINING | 0.01353 | 0.1502 | 0.1245 | 0.1213 |
| REMOVE REVIN | 0.02651 | 0.0661 | 0.2601 | 0.3266 |
| REMOVE BOTH | 0.03619 | 0.2952 | 0.4049 | 1.5892 |

Table 8: RMSE comparison between MPP-pretrained models and dedicated baselines on shallow water equations (SWE), 2D Diffusion-Reaction (DiffRe2D). Complex parameters counted as two real.

| MODEL | #PARAM | SWE | DIFFRE2D |
|---|---|---|---|
| MPP-AViT-Ti | 7.6M | 6.9E-3 | 1.1E-3 |
| UNET | 7.7M | 8.6E-2 | 6.1E-2 |
| FNO | 927K | 4.5E-3 | 8.1E-3 |
| AViT-B | | | |
| TASK-SPECIFIC | 116M | 4.9E-4 | 8.8E-4 |
| MPP | 116M | 3.6E-3 | 6.6E-4 |
| MPP + FINETUNED | 116M | 4.2E-4 | 6.2E-4 |
| MPP-AViT-S | 29M | 4.0E-3 | 7.8E-4 |
| MPP-AViT-L | 409M | 2.3E-3 | 5.0E-4 |

absorbing boundaries are complicated for numerical solvers, it seems as though attention should be able to simply not attend to waves passing out of the domain. The interaction of boundary conditions with attention therefore seems to be an important direction for future study.

### D.2 Normalization Ablations

We perform several ablations to record the impact of our proposed normalization and loss balancing approaches to the task of multiple physics pretraining. These results are shown in Table 7 using AViT-B as the baseline. As we can see both modifications greatly improve the ability of our base model to learn in this setting. Note that all of these experiments are performed in the multiple-physics settings. Results for dedicated models are listed in the main text.

Note that while training on a normalized loss appears to be important for MPP, the ordinal value of the final results is agnostic to the metric chosen. For example, we include Table 8, a version of Table 1 which reports loss in RMSE instead of NRMSE on the smaller datasets to demonstrate that while not all reported models are trained using the same metric, this does not significantly change the results. Note that not all sources report RMSE so this table contains fewer results compared to the main text.

### D.3 Broader Usage of Pretrained Representations

One of the fascinating aspects of large pretrained models is the utility of their learned features for entirely new types of prediction problems. We explore this behavior by comparing the ability of a pretrained MPP-AViT-B model to one trained from scratch to solve the inverse problem of parameter estimation for two parameters.

**Forcing Identification for Incompressible Navier-Stokes** The two sources of variation in the Incompressible Navier-Stokes simulations (Equation 7) are the initial conditions and the spatially varying forcing $f$ applied to the velocity evolution at each step. We compare the performance between the pretrained the constant forcing term used in the incompressible Navier-Stokes simulation from an input trajectory $\boldsymbol{U}_t^S$. We divide the validation set from pretraining, taking 1,000 trajectories as the new training set and using the rest for validation. Results are reported on the original test set.

**Buoyancy for Incompressible Navier-Stokes** For this, we turn to an additional fluid mechanics benchmark, PDEArena (Gupta & Brandstetter, 2022). This benchmark includes an incompressible

Table 9: RMSE for inverse problem tasks. Error from constant prediction included for context.

| TRAINING | FORCING | BUOYANCY |
|---|---|---|
| MPP | $0.20^{\pm.008}$ | $0.078^{\pm.006}$ |
| SCRATCH | $0.43^{\pm.012}$ | $0.077^{\pm.005}$ |
| MIALON ET AL. (2023) | — | $0.062^{\pm.010}$ |
| PREDICT MEAN | $1.00^{\pm.000}$ | $0.088^{\pm.000}$ |

Table 10: Per dataset NRMSE comparison for $M = 0.1$ Compressible Navier-Stokes data. R/T denote "random" and "turbulent" initial conditions from PDEBench. $\eta = \zeta$ are the bulk and sheer viscosity.

| MODEL | R-$\eta = 10^{-8}$ | R-$\eta = 10^{-2}$ | R-$\eta = 10^{-1}$ | T-$\eta = 10^{-8}$ |
|---|---|---|---|---|
| MPP-AVIT-TI | 0.0493 | 0.0274 | 0.0116 | 0.0339 |
| UNET | 0.66– | 0.71– | 5.1— | 0.19– |
| FNO | 0.28– | 0.17– | 0.36– | 0.16– |
| MPP-AVIT-S | 0.0335 | 0.0176 | **0.0071** | 0.0217 |
| FNO-B | 0.1810 | 0.1129 | 0.3800 | 0.1175 |
| MPP-AVIT-B | 0.0286 | 0.0162 | 0.0078 | 0.0169 |
| MPP-AVIT-L | **0.0234** | **0.0145** | 0.0099 | **0.0136** |

Navier-Stokes simulation with variable buoyancy ($b$ from Equation 13). Since this set was not used during training, we take 1,000 randomly sampled trajectories for train, 100 for validation, and a further 1,000 for testing. Since we are now predicting a scalar, we train a linear probe on top of the final hidden representation consisting of global average pooling and a linear head.

We observe mixed results (Table 9). Pretraining reduces the error in the forcing task by nearly half, but shows no improvement over training from scratch in the scalar prediction. Prior work (Mialon et al., 2023) was able to achieve better performance on buoyancy through Lie-transformation based contrastive pretraining using a convolutional architecture. MPP does not seem to hurt performance on this task, as the AViT trained from scratch also barely outperforms a mean prediction. However, we would expect the scalar prediction task to be easier. It is plausible that the dense prediction pretraining task is not well-suited for scalar inference or that the pooled representation frequently used for finetuning classification models in vision is not well suited to parameter inference, but the comparison of performance on this non-generative task also echoes prior work in NLP (Wang et al., 2022b) where autoregressive training has underperformed on non-generative tasks.

## D.4  Exp1: Expanded Results

Here we break out the Compressible Navier-Stokes (CNS) results from Table 1. Table 1 shows the comparison between our pretrained models and task-specific baselines; however, due to space limitations the CNS was aggregated by mach number in the main text, so we share the full CNS results here. M0.1 can be seen in Table 10. M1.0 can be seen in Table 11. Note that while it is conventional to describe these simulations in terms of dimensionless numbers like the Reynolds number, these simulations are performed at relatively low resolution, so it is likely they incur significant numerical diffusion. Thus we report the results in terms of the nominal diffusion coefficients without making claims about the Reynolds numbers of the simulation.

In examining the full CNS data, one interesting result jumps out - the most viscous systems $\eta = .1$ seem to perform relatively worse with scale. For both subsets, S was the top performing model at the highest viscosity. All other viscosities seem to benefit from scale. This does seem to have a limit, however, as Ti again loses performance. It is also important to remember that these results occur during multi-task training, so they cannot be directly interpreted in the single-task setting.

Table 11: Per dataset NRMSE comparison for $M = 1.0$ Compressible Navier-Stokes simulations. R/T denote "random" and "turbulent" initial conditions from PDEBench. $\eta = \zeta$ are the bulk and sheer viscocity.

| MODEL | R-$\eta = 10^{-8}$ | R-$\eta = 10^{-2}$ | R-$\eta = 10^{-1}$ | T-$\eta = 10^{-8}$ |
|---|---|---|---|---|
| MPP-AVIT-TI | 0.0615 | 0.0327 | 0.0171 | 0.0594 |
| UNET | 0.47– | 0.36– | 0.92– | 0.14– |
| FNO | 0.35– | 0.096- | 0.098– | 0.13– |
| MPP-AVIT-S | 0.0451 | 0.0223 | **0.0108** | 0.0425 |
| FNO-B | 0.3216 | 0.0573 | 0.0914 | 0.1100 |
| MPP-AVIT-B | 0.0386 | 0.0195 | 0.0119 | 0.0365 |
| MPP-AVIT-L | **0.0314** | **0.0171** | 0.0132 | **0.0282** |

Table 12: Test NRMSE for "Near" Compressible Navier-Stokes M0.1, $\eta = .01$.

| MODEL | # TRAINING SAMPLES (NRMSE $\times 10^{-1}$) | | | | | | | | | |
|---|---|---|---|---|---|---|---|---|---|---|
| | 100 | | 200 | | 400 | | 600 | | 800 | |
| | T+1 | T+5 | T+1 | T+5 | T+1 | T+5 | T+1 | T+5 | T+1 | T+5 |
| VIDEOMAE (K400) | 1.26 | 1.98 | 0.78 | 1.25 | 0.49 | 0.83 | 0.39 | 0.62 | 0.33 | 0.50 |
| VIDEOMAE (SSV2) | 0.95 | 1.61 | 0.63 | 1.04 | 0.42 | 0.66 | 0.33 | 0.52 | 0.25 | 0.39 |
| MPP-AVIT-B | 0.66 | 1.13 | 0.42 | 0.81 | 0.27 | 0.55 | 0.22 | 0.35 | 0.19 | 0.30 |

## D.5 Exp2: Numerical Results

We provide numerical results corresponding to Figure 3 in Tables 12 and 13. We refer to Sect. 5.2 for discussion.

## D.6 Pretraining Trajectories

Here we show example trajectories from pretrained models. Videos are included in the attached supplementary material. After pretraining, we find that the model initially produces strong predictions, but patch artifacts creep in over time.

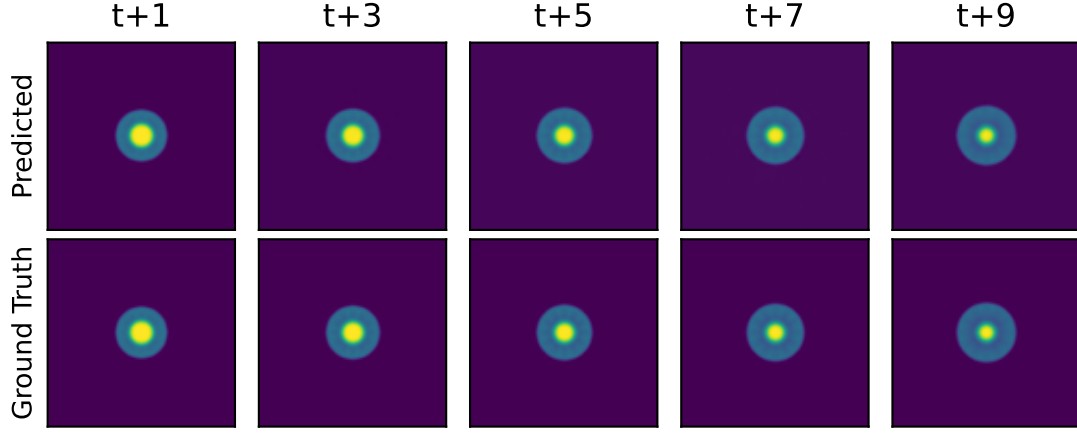

Dynamics: swe, Field h

Table 13: Test NRMSE for "Far" Compressible Navier-Stokes

| MODEL | # TRAINING SAMPLES (NRMSE $\times 10^{-1}$) | | | | | | | | | |
| | 100 | | 200 | | 400 | | 600 | | 800 | |
| | T+1 | T+5 | T+1 | T+5 | T+1 | T+5 | T+1 | T+5 | T+1 | T+5 |
|---|---|---|---|---|---|---|---|---|---|---|
| VIDEOMAE (K400) | 1.16 | 1.60 | 0.79 | 1.10 | 0.73 | 0.96 | 0.53 | 0.70 | 0.49 | 0.65 |
| VIDEOMAE (SSV2) | 0.98 | 1.42 | 0.75 | 1.03 | 0.62 | 0.84 | 0.55 | 0.74 | 0.51 | 0.67 |
| MPP-AVIT-B | 0.60 | 1.15 | 0.37 | 0.77 | 0.27 | 0.66 | .32 | 0.63 | 0.24 | 0.48 |

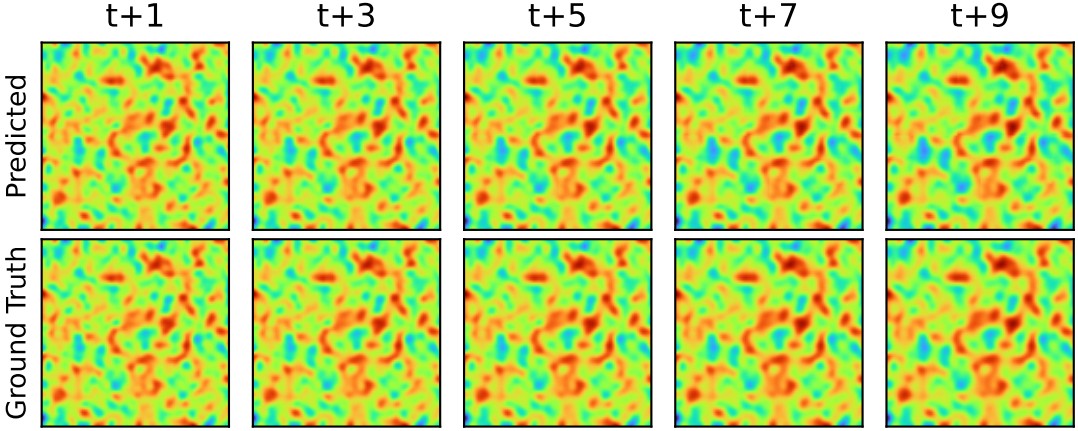

Dynamics: diffre2d, Field u

## D.7 Finetuning Trajectories

After finetuning, we find that the patch-based instability mostly disappears. Again, videos displaying longer trajectories are available in the supplementary material.

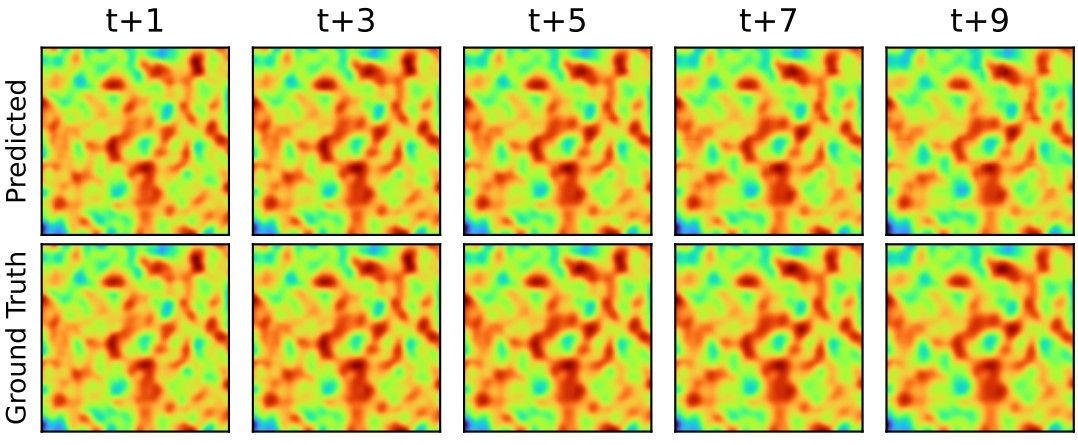

Dynamics: diffre2d, Field v

| | t+1 | t+2 | t+3 | t+4 | t+5 |

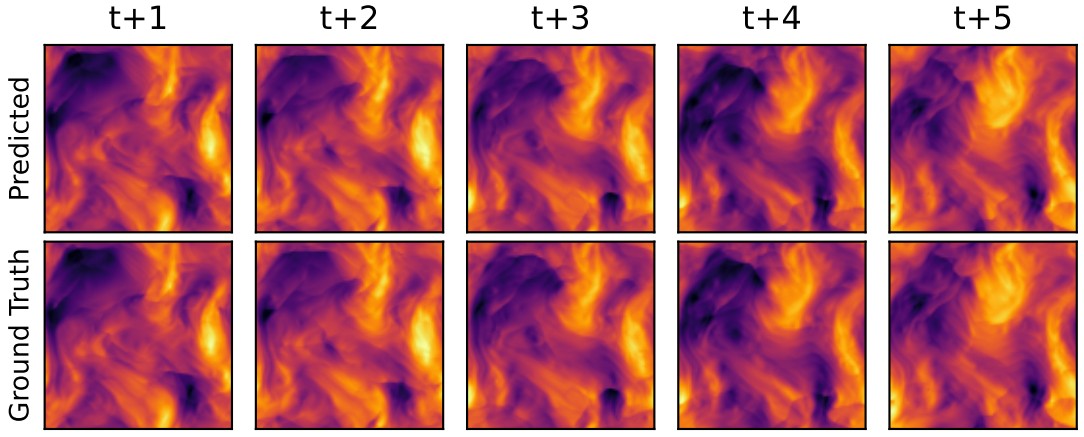

Dynamics: CNS, Field Vx

| | t+1 | t+2 | t+3 | t+4 | t+5 |

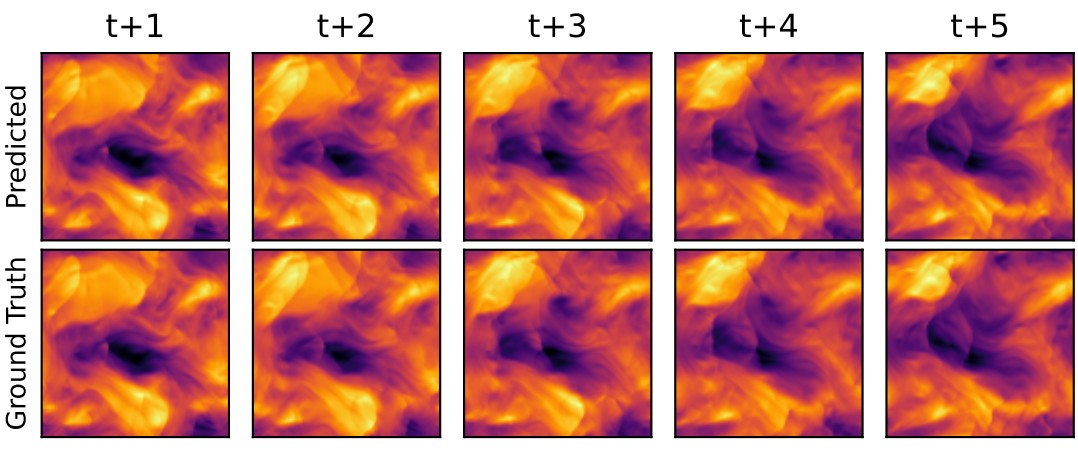

Dynamics: CNS, Field Vy

| | t+1 | t+2 | t+3 | t+4 | t+5 |

Dynamics: CNS, Field density

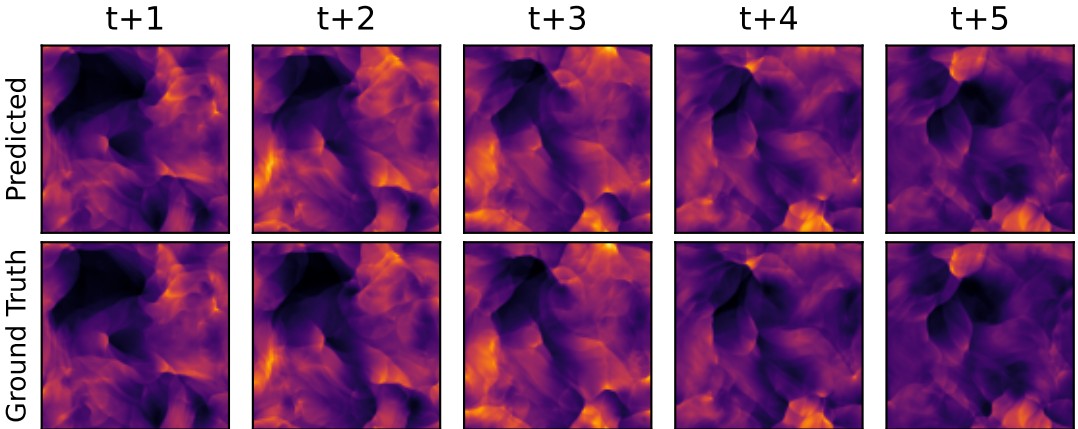

Dynamics: CNS, Field pressure

