# OpenReview forum: "Multiple Physics Pretraining for Spatiotemporal Surrogate Models"
_NeurIPS.cc/2024/Conference — NeurIPS 2024 poster_

### Official Review · Reviewer_cf8J · 2024-07-03

**Soundness:** 3
**Presentation:** 3
**Contribution:** 3
**Rating:** 7
**Confidence:** 4

**Summary:**

This paper introduces the Multiple Physics Pretraining (MPP) model, a pretraining approach for physical surrogate modeling of spatiotemporal systems using transformers. MPP uses a backbone model to predict the dynamics of several heterogeneous physical systems simultaneously. The authors include a shared embedding and normalization strategy to facilitate effective learning and transfer across different scales and magnitudes. The accuracy of MPP is validated on a fluid mechanics-oriented benchmark, showing that a MPP can perform better than the baselines without finetuning. For downstream tasks, MPP-trained models show more accurate predictions compared to training from scratch or finetuning pretrained video foundation models.

**Strengths:**

* The model is robust and outperforms task-specific baselines without finetuning.
* The authors provide open-source code and pretrained models for reproducibility.
* The method has good transfer capabilities to systems with limited training data.

**Weaknesses:**

* The network architecture is heavily oriented to 2D fluid mechanics examples, but the applicability to other domains and physics systems is not clear.
* The model is validated only with good quality data, which might not be the case for real-world measured datasets.

**Questions:**

* Line 4-5: The paper claims to "predict the dynamics of multiple heterogeneous physical systems". However, the method is only focused on 2D fluid mechanics examples. How generalizable is this approach to fully heterogeneous physics data (say, mixing electromagnetism + solid mechanics + fluid mechanics problems)?. In that case, the AViT might not a viable option anymore.
* Section 4.1: Related to the previous question. This paper relies heavily on the compositonality assumption. Indeed, all the examples are particular cases of the full compressible NS equations together with a transport equation. What if the compositionality assumption does not hold? Often in physics there are certain kinds of phenomena which might not compose trivially from the equations (for example, fluid-structure interaction, contact, etc).
* Line 167: Apart from the multihead attention from the AViT, have the authors considered a mixture of experts architecture? Would it increase the performance with respect to a single common backbone network?

Final comment: Despite the limited applicability to fluid mechanics examples, this paper contributes significantly towards the direction of foundation models for science. The methodology is novel, the results outperform baseline models and the paper is well written and structured. Based on the comments above I suggest an accept.

**Limitations:**

Limitations are addressed appropriately.

---

> ### Author Rebuttal · Authors · 2024-08-07
>
> We thank the reviewer for their time and expertise. The reviewer raised a number of interesting points that we'd like to address. Some of our responses are more discussion than rebuttal, but we hope we can ease the reviewer's concerns in certain areas and allow them to feel confident in their review.
>
> ___
> ### W1 - 2D Fluids
>
> This is a point where we believe we can offer clarification. The AViT architecture itself is designed to easily generalize to varying dimensions - for instance in our 3D inflation experiments, the axial architecture simply repeats the spatial attention operation in the added experiments. No new weights need to be added to the “inner” blocks of the network. Similarly, one could run on 1D data by collapsing a dimension of the encoder/decoder and applying spatial attention along the one dimension. However, we agree that it is currently designed for uniformly gridded meshes and that architecture changes may be required to address non-uniform grids in future work.
>
> ___
> ### W2 - Data quality
>
> This is an excellent point. We will update our limitations section to reflect this. While this paper uses data entirely from an established benchmark in order to ground our results with a novel approach firmly in the literature, we believe exploring transfer to experimentally obtained data would be an interesting and valuable study.
>
> ---
> ### Q1/2 - Generalization
>
> We feel these points are both fundamentally about generalization, so we'll split them up slightly differently and discuss generalization in terms of architecture and in terms of model capacity.
>
> First, architecture: as we mentioned in W1, we’d clarify that the AViT’s limitation is non-uniform grids rather than being strictly designed for 2D. Our inflation experiments for instance show multiple physics pretraining, even in 2D, provides value for 3D downstream tasks as well. The axis-aligned architecture, however, does require either uniformly gridded data or data that can be projected onto a uniform grid. While we believe the use of data-dependent spatial mixing (as found in attention or recent selective state space models) is important for multiple physics training, MPP does not require an AViT specifically.
>
> The question of generalizability in the sense of model capacity is more complex and likely requires downstream study. If we imagine that we were generating training data by solving random PDEs, it is intuitive that we would eventually reach a capacity limit - we’re not going to learn an infinite set of mappings from a finite set of parameters. However, we’re not generating random PDEs. Most systems of interest to researchers are either derived from conservation laws or are aggregated from particle/density level relations which leads to the very interesting question of where exactly these capacity limits are. As the data ecosystem for scientific machine learning matures, we believe these could become rich, active research areas. While we don’t answer these questions in this work, we feel that enabling the community to explore these questions is a positive rather than a negative of our work.
>
> ---
> ### Q3 - MoE options
>
> There are some appealing conceptual arguments - under the compositionality hypothesis, fields could hypothetically be routed towards modules that implement previously learned physical components. However, while MoE seems like a high potential area of exploration as models scale larger, as the dense models were working quite well, we left such explorations for future work.
>
> ---
> ### Conclusion
>
> We greatly appreciate your time and effort both in your initial review and in reading our responses. We enjoyed considering your questions and hope our responses are interesting and informative.

---

> > ### Comment · Reviewer_cf8J · 2024-08-12
> >
> > I thank the authors for the rebuttal, I have no more concerns.

---

### Official Review · Reviewer_mqdr · 2024-07-12

**Soundness:** 3
**Presentation:** 3
**Contribution:** 3
**Rating:** 5
**Confidence:** 4

**Summary:**

This paper introduces a multiple physics pretraining approach for surrogate modeling, which learns general useful features across diverse physical tasks with a shared embedding and normalization strategy. The experiment results show the proposed MPP-pretrained model outperforms task-specific baselines on all pretraining sub-tasks and also show superior finetuning results on new physics tasks.

**Strengths:**

The idea of constructing a large pre-trained base model for physical simulations is promising. The experiments are performed for diverse physical systems. The results show that large surrogate models outperform strong baselines, and even models with a relatively few parameters can learn such diverse physics evolutions and perform competitively.

**Weaknesses:**

- The validation of the approach is primarily conducted on fluid mechanics-oriented benchmarks. While this is a solid start, the applicability of the approach to other domains of physics remains to be demonstrated extensively.
- The proposed model can handle simulations on structured meshes. However, simulations on unstructured mesh are under exploration.

**Questions:**

1. Does MPP need to be trained in a context of very similar physical backgrounds (e.g., SWE, DiffRe2D and CNS)? How can we determine the similarity of multiple physics fields and whether they can be learned simultaneously?
2. Single MPP can learn the dynamics for multiple classes of physical behavior. If the physical equations are vastly different and include different forms of dynamic processes, can one model still perform well? For example, can a single model handle Newtonian fluids and non-Newtonian fluids, rigid body dynamics or elastodynamics?
3. Do baseline methods use the same normalized training loss as MPP? Why do you evaluate using Normalized Mean Squared Error (NMSE) instead of Mean Squared Error (MSE) during evaluation?

**Limitations:**

Yes

---

> ### Author Rebuttal · Authors · 2024-08-07
>
> To begin, we'd like to thank the reviewer for the effort they put forth in reviewing our work. Their feedback will help us strengthen the submission and we are grateful for their expertise. We hope that we can address their concerns through our rebuttal. If we are able to address any of your concerns, we would ask that you consider raising your evaluation accordingly.
>
> ---
> ### W1 - ...primarily conducted on fluid mechanics-oriented benchmarks...
>
> This is a great point which we mention in our limitations section. However, we'd emphasize that limited data scope can also be useful. MPP is a large change from established approaches for training surrogate models. It is therefore valuable to ground the work in the existing literature rather than creating an entirely new set of tests and comparisons - the latter invites the question of tuning effort and specialization which detracts from the goal of the paper.
>
> PDEBench, despite the restriction to fluids, is one of the more diverse spatiotemporal physics benchmarks published to date incorporating multiple equations, parameters, boundary conditions, and dimensionalities; and with ~125 citations has seen relatively wide adoption. As we state in the limitations, we believe training true "foundation models" for this space will require more data maturity, but PDEBench serves as ideal testbed to explore the question of partially overlapping physics while sticking with battle-tested benchmarks.
>
> ---
> ### W2 - Unstructured Meshes
>
> We agree that processing unstructured meshes is vital for many engineering applications, but as an active research area in its own right, irregular mesh handling is orthogonal in many senses to the questions we wanted to answer in this work. In this paper, we wanted to answer two fundamental questions - first whether it is possible for individual models to learn multiple sets of dynamics. Secondly - whether this is beneficial for transfer to new systems whose physics partially overlap with the training data. Uniform grids were sufficient for answering both of these emphatically in the affirmative. Extending our approach to non-uniform grids opens up some really exciting research directions related to transfer across geometry, but we felt the initial exploration would benefit from tighter scope and reduced degrees of freedom.
>
> ---
> ### Q2/Q1 - Major differences in physical process and the limits of transfer
>
> These are very interesting questions that highlight the type of research questions opened up by MPP. Large scale pretraining in vision and language have opened up new avenues for interpretability research and we believe MPP could do the same for data-driven spatiotemporal physics. In this paper, we're focused on partially overlapping physics on a fluid-dominated dataset. PDEBench is clearly not the whole of physics, but as one can see comparing INS and CNS, the restricted class covered is still quite diverse. Demonstrating transfer over this range is a significant advancement over prior work and an important step forward in the development of foundation models for this space.
>
> Ultimately exploring the limits of transfer will require more mature datasets and significant amounts of compute. Identifying a priori approaches for quantitatively evaluating whether dynamics are similar or finding where model capacity begins to breakdown are natural and interesting research questions that emerge once MPP is established as a possibility. While there is more to explore, we feel opening these new exciting directions is a strength of our work rather than a weakness.
>
> ---
> ### Q3 -
>
> NMSE was chosen primarily for consistency with the existing literature. It is the only metric reported across all papers we drew baselines from. However, we also agree with this convention. Normalized metrics are viewed as more interpretable. In a vacuum, it can be difficult to tell if an RMSE of $10^{-6}$ is “accurate” since the underlying field may have values on order $10^{100}$ or $10^{-14}$. NRMSE avoids that issue, though the learning task itself can still vary in difficulty.
>
> Since all baselines are drawn from independent studies, they are also trained using the procedures chosen by the original authors. The PDEBench baselines are trained using RMSE while ORCA is trained with NRMSE. For single task models, there is very little difference here as PDEBench fields have similar norms across samples within a given dataset. The exception to this are the NS where there is variation within fields so the relative weighting of fields in the aggregate loss is different. However, this has little effect on aggregate metrics in practice - the ordinal rankings do not change in RMSE. We understand the concern and will add an RMSE table to the appendix to address it.
>
> ---
> ### Conclusion
>
> Once again, we'd like to thank the reviewer for the energy they put into helping us refine our paper. If we were able to address any of your concerns, we'd ask that you raise your score proportionally.

---

> > ### Comment · Reviewer_mqdr · 2024-08-12
> >
> > I have read the rebuttal and appreciate the authors' efforts in addressing my concerns.

---

### Official Review · Reviewer_1X1Y · 2024-07-15

**Soundness:** 4
**Presentation:** 4
**Contribution:** 4
**Rating:** 10
**Confidence:** 5

**Summary:**

This article introduces a transformer model which turns across multiple special temporal physics. Using a clever choice of standardization and scaled training, they managed to train a model that can predict the next step given context of snapshots. In the article, they show that a single model can learn dynamics from multiple classes of physical behavior, and a model can learn partially overlapping physics for transfer learning. In the experiment, they show competitive results across multiple physics with a strong emphasis on the challenging fluid dynamics. They also show capacity to transfer efficiently to low data regime. They also show promising results of using their 2-D model for 3-D solves.

**Strengths:**

This is an outstanding article and I have only a few minor comments.

**Weaknesses:**

Please refer to the minor comments in Questions.

**Questions:**

Could you please address the following minor comments?
- Please provide details of the fine tuning with MPP, I could only find details for videoMAE.
- Please provide details on how to predict the first T_S first snapshots since they have incomplete context.
- Could you please give an intuition why MPP performs so much better than the rest in figure 3b?
- Please discuss whether the learned tokens from multiple physics could be interpretable or how could this MPP be leveraged to make the predictive model more trustworthy?
- L. 146. The word “function” is missing after “scalar”
- L. 253 it should be T_S instead of T^S
- L. 350 what does “Ti” scale mean? Is it before De-RevIn?

**Limitations:**

Limitations are adequately addressed.

---

> ### Author Rebuttal · Authors · 2024-08-07
>
> We'd like to thank the reviewer for their thorough reading of our paper. The reviewer raises some great questions and discussion topics which we aim to address below.
>
> #### Q1 -
>
> The finetuning settings are provided in C.3 for 2D and C.4 for 3D. However, while the instructions for initializing new field embeddings for new fields are contained in the repository, those details are not currently in the paper. In addition to what is stated in C.3/4, for new fields, one must also expand the field embedding weights and randomly initialize the new embeddings.
>
> We will add this and expand the finetuning details in a manner more similar to what is described for VideoMAE to centralize the information.
>
> #### Q2 -
>
> The initial snapshots must come from the data source. It is possible that the context length can be reduced or use variable context lengths with some minor data augmentation during training - 16 was largely chosen for an equitable comparison to VideoMAE - but ultimately some history is needed. This is because the model needs to use a form of in-context learning to differentiate between similar dynamics. For instance, if we’re looking at the PDEBench compressible Navier-Stokes data, we often have similar initial conditions in systems with slightly different viscosities. In a single snapshot, the two cannot be disentangled, but over a moderate length trajectory, one can observe the dynamics evolving differently.
>
> In designing models for this space, the model either needs to know the exact system coefficients or be able to implicitly infer dynamics from a provided histor. We felt the history approach was more in line with potential workflows. Given observational data, for instance, it is unlikely a user could provide the Reynolds number, but they can almost certainly take a small number of measurements separated in time. Or in a design optimization setting where one might want to approximate complex non-differentiable legacy code with the model with a differentiable surrogate, it might be trivial to generate a short history, but several months of effort to identify all of the conditional modeling assumptions in the code base.
>
> #### Q3 -
>
> Thanks for raising this point! This is something we shied away from discussing explicitly in the paper as it requires some level of PDE background, but since it has come up with several reviews, we will add this information to the paper itself.
>
> In short, there is a degree to which 3B can be seen as a more realistic instance of what we see in our motivating example of section 4. In Section 4, we explore whether learning partially overlapping physics can be beneficial in the extremely simplified setting of 1D linear PDEs on periodic domains. Our pretraining data contains two families of fluid flow - viscous incompressible Navier-Stokes (INS) and the inviscid shallow water equations (SWE). The INS data can be seen as globally parabolic. Even locally, there is no point where viscous forces will allow for shock formation. The SWE simulations have locally hyperbolic behavior meaning we can see discontinuities (shocks) form naturally in the flow.
>
> 3A incorporates new physics, but with high viscosity and low mach number, qualitatively it should be similar to the INS pretraining data. 3B is nearly inviscid and occurs at high mach number. This allows for local behavior resembling hyperbolic systems to emerge. The model never saw this in momentum-driven transport problems (INS), but it has seen SWE which has hyperbolic behavior but does not provide the velocity fields for computing transport. So like in Section 4, we have two sources of training data that contain sub-components of the new physical process, but finetuning is required since the model has never seen those components together.
>
> #### Q4 -
>
> This is a very interesting question. It's not clear that this approach improves interpretability, though analyzing MPP with mechanistic interpretability tools to identify what type of algorithm the model is employing to make the forecast is a direction we're excited about for future work. However, there is a strong case to be made for reliability. Imagine we train a transformer on a single dataset. In this dataset, fluids move only a single cell over each timestep. The model would predictably learn to move based on a sign function on velocity - this isn't the correct function, but its a function that is simple to learn and fits the data perfectly. MPP enables the model to learn from multiple datasets simultaneously ensuring coverage for a wide variety of data. This makes it difficult for models to learn trivial idiosyncrasies of single datasets as the model must learn compressed representations relevant to many datasets simultaneously.
>
> #### Quick Qs
> - Q5 - Thanks! We will fix this.
> - Q6 - Thank you! Fixed.
> - Q7 - This is the model size "tiny". It corresponds to the smallest AViT in Table 1
>
> #### Conclusion
>
> Again, we'd like to thank the reviewer for their efforts. We hope this discussion is interesting and provides useful information!

---

> > ### Comment · Reviewer_1X1Y · 2024-08-09
> >
> > I thank the authors for addressing my questions in a satisfactory way.

---

### Official Review · Reviewer_1VF5 · 2024-07-29

**Soundness:** 3
**Presentation:** 3
**Contribution:** 2
**Rating:** 4
**Confidence:** 4

**Summary:**

The work proposes the idea of pre-training models on multiple PDE problems and demonstrates that such a pretrained model can be effectively fine-tuned on a target PDE when the pre-training and fine-tuning PDEs are similar. The authors carefully constructed the scalable transformer architecture by employing an axial attention mechanism of high-resolution PDE datasets. They validated their hypothesis through empirical analysis of the 2D Shallow Water equation, Diffusion-Reaction equation, and Navier-Stokes equation in both 2D and 3D domains.

**Strengths:**

1. The work tackles a contemporary yet unresolved issue of designing a foundation model for solving PDEs.

2. The proposed model showed impressive performance on the selected PDE problems.

**Weaknesses:**

1. **Only transferable to very similar PDEs**.
The argument for pre-taining on multiple PDEs was to generalize to unknown PDEs in low data regimes. However, the experiment design lacks practicality. The demonstrated example of finetuning on CNS while pre-training on a large amount of INS data lacks the similarity of any practical situation. For example, a demonstration of the effectiveness of the model on the introduction of new variables, change of domain or geometry, or a moderately different PDE (such as keeping out SWE from the pretraining dataset and finetuning on a small amount of SWE data) would have been a more practical choice.

2. The domain expansion from 2D to 3D adaptation experiment was intriguing. However, the performance of AVIT-SCRATCH and AVIT-MPP is quite similar. This raises the question of whether the benefit comes from MPP or the designed architecture. Nonetheless, this does not align with the primary message of the paper.

**Questions:**

1. The model does not consider any PDE-specific input. If the pre-training dataset includes data from very similar PDEs (such as Navier-Stokes equations with nearly identical viscosities), would the model be able to distinguish between them effectively?
2. In Table 1, multiple bold elements appear in the same column. What do they signify?
3. How well does the proposed architecture handle changes in spatial and temporal resolution during testing?

**Limitations:**

yes

---

> ### Author Rebuttal · Authors · 2024-08-07
>
> First off, we’d like to thank the reviewer for their time and effort in reviewing our paper. Your feedback will make the paper stronger. There are a few points raised in the review that we hope to clarify.
>
> ---
> ## W1 - Only transferable to very similar PDEs.
>
> The hypothesis we explore in this work is that “learning partially overlapping physics is beneficial for transfer learning”. This is what we believe our work has shown, though the level of heterogeneity is quite a bit larger than what may be obvious from the equations themselves.
>
> CNS and INS are separated by the Mach number. Low Mach CNS simulations, as in the dataset we call "Near" are qualitatively very similar to INS simulations, but this is no longer true at high Mach number. In PDEBench specifically, there are even more significant differences. __The CNS data in PDEBench possesses both previously unseen geometries and new variables when compared to the pretraining data in Section 5.2.__ We’ll break these differences into several sections: geometry, physics and state variables, and the significance of the experiment.
>
> .
> ### Geometry
> The INS data uses no-slip boundary conditions - these approximate a domain with solid walls. The CNS data uses periodic boundary conditions which imply no walls. Somewhat unusually since periodic boundaries are typically less expensive to simulate, in our transfer experiments, the model has actually never seen periodic geometry. Without walls, the small-scale behavior in the flow is a product of colliding shocks rather than the forcing and wall dynamics from the INS simulations.
>
> .
> ### Physics and Variable Comparison
> The difference in settings also results in different represented fields. The INS setting in PDEBench is a closed, inhomogeneous, highly viscous (coefficient=$10^{-2}$) flow driven by a forcing term. It is a domain without a pressure gradient and where any density variation is driven by the presence of immersed particles in the flow rather than density differences due to compression or expansion. It is fully characterized by *velocity*, *forcing*, and the *density of the immersed particles*. The behavior can locally be classified as parabolic everywhere - viscous forces damp any emerging shocks such that even discontinuous initial conditions will become smooth.
>
> The CNS physics, particularly for “Far” which has viscosity coefficients of $10^{-8}$, are very different. They are unforced, the density of the fluid itself varies freely due to compressible behavior, and pressure is not constant. Thus these simulations add fluid density and pressure fields that were also never seen during pretraining. In terms of the physics, near-inviscid flow with Mach number of 1 allows for local behavior resembling hyperbolic systems - shock formation and transport.
>
> .
> ### Significance
> What’s really exciting to us, and we hope for readers as well, is that this experiment reinforces the compositionality hypothesis: the model has seen hyperbolic and momentum-driven transport during training, but never in the same system. The inviscid shallow water equations used to generate the data are hyperbolic and shock forming while the model had seen momentum-driven transport in INS. Thus this transfer experiment can be interpreted as a more complex version of our motivating example in Section 4.
>
> This is also one of the most practically useful examples of transfer. Incompressible flow solvers tend to be significantly cheaper to run than compressible flow, so learning from incompressible data would provide an enormous advantage for data acquisition.
>
> ---
> ## W2 - 3D transfer
>
> We’re glad that you found the adaptation experiment interesting. 2D-3D transfer is an area that we're excited about as well due to the scaling benefits. Most of the 2D data in PDEBench can be generated in less than 24 hours on current hardware. Equivalent resolution 3D data can take weeks or months on similar hardware. From a ML training perspective, 512 x 512 input is straightforward. Handling 512 x 512 x 512 requires specialized distribution strategies. The ability to use 2D data can be an enormous cost savings for both training data generation and pretraining.
>
> While “quite similar” is a matter of opinion, we’d note that these are 4.1% and 11.7% improvements. It looks small because the AViT itself is a significant improvement over the PDEBench baselines. We felt it was important to contextualize the results with the stronger AViT comparison to more accurately demonstrate the impact which is greater than 10% for the smaller dataset where transfer is more vital.
>
> ---
> ## Q1 - ... would the model be able to distinguish between them effectively
>
> This depends on the level of similarity, but in general it is possible and demonstrated in our experiments. The PDEBench CNS data uses a range of viscosity parameters. In Tables 9/10 in the supplementary material, we can see that the pretrained models trained on all datasets/parameters are still able to outperform the PDEBench baselines on each dataset.
>
> ---
> ## Q2 - Bolded tables
>
> Bolding indicates top performance within a parameter count bracket. “L” is the top performer overall showing that we see significant benefits from scale.
>
> ---
> ## Q3 - Varying resolution
>
> In general, the model needs to have seen problems in a similar range - we would not expect zero shot transfer to unseen discretizations. Using 1D example from section 4, we can intuit that if the model has always been trained at velocities such that density moves exactly one cell during training, the model will learn to only explore neighboring cells, and decreasing the temporal resolution will result in nonsensical zero-shot answers. This is actually one of the perks of MPP - the data coverage from using MPP helps avoid overfitting due to data idiosyncrasies.
>
> ---
> ## Conclusion
>
> Again, we appreciate the reviewer's time and effort. If we were able to alleviate your concerns, we'd ask that you adjust your evaluation to reflect it.

---

> > ### Comment · Reviewer_1VF5 · 2024-08-12
> >
> > Thank you for explaining.
> >
> > I believe the concept of "new physics" depends on interpretation. Some may argue that all Navier-Stokes partial differential equations (NS pde) with different physical parameters exhibit the same physics because they are governed by the same equation. On the other hand, some may argue that NS equations with different parameters represent different physics.
> >
> > I strongly advise the author to clarify the nature and scope of the "new physics" being addressed in the abstract and the contribution.

---

> > > ### Author Response · Authors · 2024-08-13
> > >
> > > Thank you for the suggestion. That's a good point. Our discussion here shows there is room for disagreement about what constitutes new physics and that the paper could benefit from more precision on that point. We will update the wording to state exactly what the new physics we're referring to are.

---

### Decision · Program_Chairs · 2024-09-25

**Decision:**

Accept (poster)

**Comment:**

The rebuttal has addressed the reviewers' concerns. Reviewer 1VF5 brought up a good point regarding the exact definition of "new physics". The main contention is whether "new physics" is limited to only the "difference" in the physical parameters or also the difference governing equations. The authors have promised to clarify and properly define this term.